# An evaluation of the heat test for the ice-nucleating ability of minerals and biological material

Martin I. Daily[1], Mark D. Tarn[1], Thomas F. Whale[1]* and Benjamin J. Murray[1]

[1]Institute of Climate and Atmospheric Science, School of Earth and Environment, University of Leeds, Leeds, LS2 9JT, UK
*Current address: Department of Chemistry, University of Warwick, Gibbet Hill, Coventry, CV4 7AL, UK

Correspondence to: Martin I. Daily (_m.i.daily1@leeds.ac.uk_) and Benjamin J. Murray (b.j.murray@leeds.ac.uk)

**Abstract.** Ice-nucleating particles (INPs) are atmospheric aerosol particles that can strongly influence the radiative properties and precipitation onset in mixed-phase clouds by triggering ice formation in supercooled cloud water droplets. The ability to distinguish between INPs of mineral and biological origin in samples collected from the environment is needed to better understand their distribution and sources. A common method for assessing the relative contributions of mineral and biogenic INPs in samples collected from the environment (e.g., aerosol, rainwater, soil) is to determine the ice-nucleating ability (INA) before and after heating, where heat is expected to denature proteins associated with some biological ice nucleants. The key assumption is that the ice nucleation sites of biological origin are denatured by heat, while those associated with mineral surfaces remain unaffected; we test this assumption here. We exposed atmospherically relevant mineral samples to wet heat (INP suspensions warmed to above 90 °C) or dry heat (dry samples heated up to 250 °C) and assessed the effects on their immersion mode INA using a droplet freezing assay. K-feldspar, thought to be the dominant mineral-based atmospheric INP type where present, was not significantly affected by wet heating, while quartz, plagioclase feldspars and Arizona test Dust (ATD) lost INA when heated in this mode. We argue that these reductions in INA in the aqueous phase result from direct alteration of the mineral particle surfaces by heat treatment rather than from biological or organic contamination. We hypothesise that degradation of active sites by dissolution of mineral surfaces is the mechanism in all cases due to the correlation between mineral INA deactivation magnitudes and their dissolution rates. Dry heating produced minor but repeatable deactivations in K-feldspar particles but was generally less likely to deactivate minerals compared to wet heating. We also heat tested biogenic INP proxy materials and found that cellulose and pollen washings were relatively resistant to wet heat. In contrast, bacterial and fungal derived ice-nucleating samples were highly sensitive to wet heat as expected, although their activity remained non-negligible after wet heating. Dry heating at 250 °C leads to deactivation of all biogenic INPs. However, the use of dry heat at 250 °C for the detection of biological INPs is limited since K-feldspar's activity is also reduced under these conditions. Future work should focus on finding a set of dry heat conditions where all biological material is deactivated, but key mineral types are not. We conclude that, while wet INP heat tests at (>90°C) have the potential to produce false positives, i.e., deactivation of a mineral INA that could be misconstrued as the presence of biogenic INPs, they are still a valid method for qualitatively detecting fungal and bacterial INPs in ambient samples if the mineral-based INA is controlled by K-feldspar.

**1 Introduction**

In the absence of nucleation sites, cloud water droplets can supercool to temperatures below around −35 °C before freezing via homogeneous nucleation (Ickes et al., 2015; Herbert et al., 2015). However, a rare subset of atmospheric aerosol particles known as ice-nucleating particles (INPs) can elevate the temperature of ice formation (Murray et al., 2012; Hoose and Möhler, 2012; Kanji et al., 2017). INPs are important because newly formed ice crystals can grow at the expense of supercooled liquid droplets. This is a process that strongly modulates the radiative properties of shallow mixed-phase clouds (i.e., their albedo) (Storelvmo and Tan, 2015; Murray et al., 2021), can initiate precipitation by enhancing collision and coalescence processes (Vergara-Temprado et al., 2018; Rosenfeld et al., 2011) and can influence anvil cirrus properties in deep convective systems (Hawker et al., 2021).

To represent the impact of INPs on clouds in our models, we must improve our understanding of the global distribution and temporal variability of INPs. However, much uncertainty remains regarding the distribution, sources, and relative ice-nucleating ability (INA) of INPs throughout the Earth's atmosphere (Kanji et al., 2017; Huang et al., 2021; Murray et al., 2021). Two important general categories of INPs are mineral dust (Hoose et al., 2010; DeMott et al., 2003; Vergara-Temprado et al., 2017) and biogenic materials (Vergara-Temprado et al., 2017; Creamean et al., 2013).

Laboratory and field data indicate that mineral dusts often dominate the INP population relevant for mixed-phase clouds below around −15 °C (O'Sullivan et al., 2014; Murray et al., 2012; Ansmann et al., 2008; Niemand et al., 2012; Ullrich et al., 2017). Potassium rich feldspars (K-feldspars) are considered to be the most important ice-nucleating mineral commonly present in airborne mineral dust (Atkinson et al., 2013; Harrison et al., 2019; Zolles et al., 2015; Augustin-Bauditz et al., 2014; Kaufmann et al., 2016), with immersion mode nucleation observed at temperatures warmer than −5 °C in laboratory experiments (Harrison et al., 2016; Kaufmann et al., 2016; Whale et al., 2017). Quartz and the other feldspar varieties, plagioclase and albite, are thought to play a lesser role than K-feldspar in terms of mineral dust INA (Harrison et al., 2019; Harrison et al., 2016; Atkinson et al., 2013), with quartz being on average the more abundant of these in the atmosphere (Murray et al., 2012).

Biological INPs are capable of nucleating ice at much warmer temperatures than all but the most active minerals, and can include primary biological particles (PBAPs) such as bacteria, fungal spores, pollen grains, fragments of terrestrial organic material such as cellulose (Hiranuma et al., 2015b; Hiranuma et al., 2019) and macromolecules of marine biogenic origin (Schnell and Vali, 1976; Warren, 1987; Wilson et al., 2015; McCluskey et al., 2018a). Atmospheric concentrations of ice-active bacteria, fungal spores and pollen grains are much smaller than mineral dusts (Hoose et al., 2010). Estimates of the mass of PBAPs emitted to the atmosphere annually range from low hundreds to ~1000 Tg (Hoose et al., 2010; Jaenicke et al 2005) compared to 1,000 - 3,000 Tg per year for mineral dust (Zender et al., 2004). However, the concentration of fragments of biogenic INPs may be much greater given the release of macromolecular INPs (Augustin et al., 2013; O′Sullivan et al., 2015) and their adsorption onto lofted soil dust (Schnell and Vali, 1976; O'Sullivan et al., 2016). Also the sources and atmospheric distribution of biogenic INPs are less well characterised compared to those of minerals dusts (Huang et al., 2021; Kanji et al., 2017), owing to the diversity of marine and terrestrial sources that may be subject to seasonal variations (Conen

et al., 2015; Schneider et al., 2020; Šantl-Temkiv et al., 2019) or influenced by anthropogenic activities such as agricultural processes (Garcia et al., 2012; Suski et al., 2018; O'Sullivan et al., 2014).

Biological INPs tend to nucleate ice at temperatures where they may initiate secondary ice production processes (Morris et al., 2014; Field et al., 2017), thus amplifying their effect in clouds. Biogenic INPs could also play an important role in feedback processes in the rapidly warming Arctic climate, as increasing surface temperatures may expose new terrestrial sources in thawing permafrost (Creamean et al., 2020), newly exposed glacial outwash sediments (Tobo et al., 2019) or reveal new marine reservoirs as the sea ice coverage is reduced (Hartmann et al., 2020). While the INA of mineral dust from various arid sources around the world is relatively similar (within around a factor of 10) (Niemand et al., 2012; Atkinson et al. 2013), the INA of biological material varies massively between the various sources, which makes predicting the INP population of biological material particularly challenging.

Much effort in the past decade has been put into not only collecting and identifying biogenic INPs or their markers in the environment, but also in determining their relative contributions to the total measured INP population (Huang et al., 2021). While techniques such as genomic sequencing (Garcia et al., 2012; Huffman et al., 2013; Hill et al., 2014; Christner et al., 2008) and microscopy (Huffman et al., 2013; Sanchez-Marroquin et al., 2021) can reveal the presence of biological species in an aerosol sample that has been found to contain INPs, it remains difficult to characterise the ice-nucleating ability of these species over other constituents (e.g. mineral dusts) when a sample's INA is analysed by, for example, a droplet freezing assay alone. To date, no high-throughput technique has been established that can directly identify both the composition and nucleation temperatures of a specific INP type within a sample. However, a widely used methodology for performing an indirect assessment of the contribution of mineral vs. biogenic INPs involves treating a collected aerosol sample (or other INP-containing media) with heat and comparing its INA spectrum before and after heating. Changes in INA can then be related to the presence and domination of biogenic INPs over inorganic INPs based on several assumptions, as discussed below. This heat treatment procedure has the advantages of being suitable for high-throughput offline sample analysis and does not require specialised equipment or the addition of reagents to selectively degrade biological material such as hydrogen peroxide (Suski et al., 2018; O'Sullivan et al., 2014; Tobo et al., 2019), lysozyme (Joyce et al., 2019; Henderson-Begg et al., 2009; Christner et al., 2008) or guanidinium hydrochloride (Conen and Yakutin, 2018). We have compiled a list of past studies which have employed heat tests to detect biological INP with the conditions and method of INP detection in Table 1.

**Table 1:** List of past studies in which heat treatments were used to infer the presence of biological INPs in samples of various environmental media.

| Study | Sample media | Heat treatment method | Ice nucleation measurement method |
|---|---|---|---|
| **Baloh et al., 2019** | Snow and surface water | Wet: 95 °C for 20 min | DFA: 50 µL droplets in 96-well plates |
| **Barry et al., 2021** | Aerosol from wildfire smoke plume | Wet: 95 °C for 20 min | DFA: 50 µL droplets in 96-well plates |
| **Boose et al., 2019** | Desert dusts from nine worldwide locations | Dry: 300 °C for 10 h | Ice crystal counting by optical particle counter downstream of CFDC* |

| Study | Sample media | Heat treatment method | Ice nucleation measurement method |
|---|---|---|---|
| **Christner et al., 2008a and b** | Snow and rainwater | Wet: 95 °C for 10 min | DFA: 0.25 - 1 mL aliquots in test tubes |
| **Conen et al., 2011** | Soils with varying organic content | Wet: 100 °C for 10 min | DFA: 50 µL droplets in microfuge tubes |
| **Conen et al., 2016** | Aerosol and leaf litter suspension | Wet: 80 °C for 10 min | DFA: unstated volume in microfuge tubes |
| **Conen et al., 2017** | Aerosol sampled on hillside | Wet: 90 °C for 10 min | DFA: filter punches immersed in 100 µL droplets in microfuge tubes |
| **Creamean et al., 2018** | Bulk seawater and sea surface microlayer | Wet: 90 °C for 30 min | DFA: 2.5 µL droplets on cooling stage |
| **Creamean et al., 2020** | Permafrost soil and ice wedge | Wet: 95 °C for 20 min | DFA: 50 µL droplets in 96-well plates |
| **D'Souza et al., 2013** | Plankton sample from frozen lake | Wet: 45, 65 and 90 °C for 2 h | DFA: 80 µL aliquots in microcapillary tubes |
| **Du et al., 2017** | Rainwater | Wet: 100 °C for 10 min | DFA: 10 µL droplets on cooling stage |
| **Garcia et al., 2012** | Aerosol and surface dust collected on a farm | Wet: 98 °C for 20 min | DFA: 30 or 50 µL droplets in 96-well plates |
| **Gong et al., 2020** | Bulk seawater and sea surface microlayer, cloud water and aerosol | Wet: 95 °C for 1 h | DFA: 1 µL droplets on cooling stage and 50 µL droplets in 96-well plates |
| **Hara et al., 2016a** | Snow collected from ground | Wet: 40 °C and 90 °C for 1 h | DFA: filter punches immersed 0.5 mL in microfuge tubes |
| **Hara et al., 2016b** | Aerosol collected on building top | Wet: 90 °C for 1 h | DFA: filter punches immersed 0.5 mL in microfuge tubes |
| **Hartmann et al., 2020** | Bulk seawater, sea surface microlayer and fog water | Wet: 95 °C for 1 h | DFAs: 1 µL droplets on cooling stage and 50 µL droplets in 96-well plates |
| **Henderson-Begg et al., 2009** | Lichen samples and aerosol sample in urban location | Wet: 37, 60 and 90 °C for unspecified duration | Not stated |
| **Hill et al., 2014** | Vegetation washings and snow and hail from ground | Wet: 60 °C and 90 °C for 10 min | DFA: 50 µL droplets in 96-well plates |
| **Hill et al., 2016** | Topsoil | Wet: 60 °C and 105 °C for 20 min | DFA: 50 µL droplets in 96-well plates |
| **Hiranuma et al., 2021** | Aerosol and surface dust sampling on a cattle farm | Wet: 100 °C for 20 min | DFAs: 50 µL droplets in 96-well plates |
| **Irish et al., 2017** | Bulk seawater and surface microlayer | Wet: 100 °C for 1 h | DFA: 0.6 µL droplets on cooling stage |
| **Iwata et al., 2019** | Aerosol collected on building in forest | Dry: 150 °C for 10 min | Visual identification of ice growing on particles on cooling Si substrate |
| **Joly et al., 2014** | Cloud water | Wet: 95 °C for 10 min | DFA: 20 µL aliquots in microfuge tubes |
| **Joyce et al., 2019** | Rainwater, sleet and snow | Wet: 95 °C for 10 min | DFA: 200 µL droplets in 96-well plates |

| Study | Sample media | Heat treatment method | Ice nucleation measurement method |
|---|---|---|---|
| **Knackstedt et al., 2018** | River water and aerosolised river water | Wet: 95 °C for 20 min | DFA: 80 µL droplets in 96-well plates |
| **Lu et al., 2016** | Rainwater | Wet: 100 °C for 20 min | DFA: 10 µL droplets on cooling stage |
| **Martin et al., 2019** | Rainwater | Wet: 90 °C for 20 min | DFA: 50 µL droplets in 96-well plates |
| **McCluskey et al., 2018a** | Aerosol at coastal site | Wet: 95 °C for 20 min | DFA: 50 µL droplets in 96-well plates; CFDC* |
| **McCluskey et al., 2018b** | Sea spray aerosol, bulk seawater and sea surface microlayer | Wet: 95 °C for 20 min | DFA: 50 µL droplets in 96-well plates |
| **Michaud et al., 2014** | Hailstones | Wet: 95 °C for 10 min | DFA: 50 µL droplets in 96-well plates |
| **Moffett et al., 2018** | River water | Wet: 90 °C for 10 min | Differential scanning calorimetry of river water emulsion |
| **Moffett et al., 2018** | River water | Wet: 95 °C for 20 min | DFA: 80-100 µL droplets in 96-well plates |
| **O'Sullivan et al., 2014** | Agricultural soils | Wet: 90 °C for 10 min | DFA: 1 µL droplets on cooling stage |
| **O′Sullivan et al., 2015** | Woodland soils | Wet: 90 °C for 45 min | DFA: 1 µL droplets on cooling stage |
| **O′Sullivan et al., 2018** | Aerosol sampling on an arable farm | Wet: 95 °C for 1 hr | DFA: 1 µL droplets on cooling stage |
| **Paramonov et al., 2018** | Soil and desert dusts | Dry: 300 °C for 2 h | Ice crystal counting by optical particle counter downstream of CFDC* |
| **Šantl-Temkiv et al., 2015** | Snow and rainwater | Wet: 95 °C for 10 min | DFA: 240 - 300 µL droplets in 96-well plates |
| **Šantl-Temkiv et al., 2019** | Aerosol and snow samples | Wet: 100 °C for 10 min | DFA: 100 - 200 µL droplets for snow samples and filter punches immersed in 50 µL droplets in 96-well plates |
| **Schneider et al., 2021** | Aerosol collected from a boreal forest | Wet: 95 °C for 20 min | DFA: 50 µL droplets in 96-well plates |
| **Schnell and Vali, 1976** | Leaf litter collected from various locations worldwide and seawater | Wet: 60 – 100 °C for unspecified duration | DFA |
| **Seifried et al., 2021** | **Aerosol collected from Alpine environment** | **Wet: 98 °C for 1 h** | **DFA: 3 µL droplets in multiwell plates** |
| **Steinke et al., 2016** | Agricultural soils | Dry: 110 °C for 1 h | Ice crystal concentration by optical particle counter in cloud chamber |
| **Suski et al., 2018** | Aerosol and surface dust sampling on an arable farm | Wet: 95°C for 20 min. Dry: 300 °C upstream of CFDC | DFA: 50 µL droplets in 96-well plates; ice crystal counting by optical particle counter downstream of CFDC* |

| Study | Sample media | Heat treatment method | Ice nucleation measurement method |
|---|---|---|---|
| **Tobo et al., 2014** | Agricultural soil dusts | Dry: 300 °C for 2 h | Ice crystal counting by optical particle counter downstream of CFDC* |
| **Tobo et al., 2020** | Aerosol collected from tall TV mast in Tokyo, Japan | Wet: 100°C for 1 h | DFA: 5 µL droplets on cooling stage |
| **Tesson and Šantl-Temkiv, 2018** | Snow | Wet: 100 °C for 10 min | DFA: droplets of unspecified volume on cooling stage |
| **Wilson et al., 2015** | Bulk seawater and sea surface microlayer | Wet: 8 temperatures between 20 °C and 100 °C for 10 min | DFA: 1 µL droplets on cooling stage |
| **Yadav et al., 2019** | Rainwater and desert dust from surface | Wet: 100 °C for 10 min | DFA: 1 µL droplets on cooling stage |
| **Zinke et al., 2021** | Cloud water | Wet: 100 °C for 30 min | DFA: 1 µL droplets on cooling stage |

CFDC = Continuous Flow Diffusion Chamber

DFA = Droplet freezing assay

The identification of the presence of biogenic INPs using a heat test is based on the assumption that heat will inactivate biogenic (often but not always explicitly proteinaceous) INPs, yielding a reduction in ice nucleation temperatures following the treatment, while the INA of inorganic INPs (likely to be dominated by mineral dust) will remain unaffected (Conen et al., 2011). In addition to merely determining the presence of biogenic INPs, this method has also been used by some researchers to quantify the abundance of biogenic INPs in their samples by evaluating the magnitude of the INA reduction (Christner et al., 2008; Joly et al., 2014; Joyce et al., 2019). The assumption that protein-bearing biological INPs associated with bacteria and fungi can lose at least some of their INA when sufficiently heated (up to 100 °C) has been confirmed via many previous studies (Lundheim, 2002; Pummer et al., 2015; Roy et al., 2021) (see the review of Lundheim (2002) for an overview). However, other types of biogenic INPs, for example macromolecules derived from pollen (Pummer et al., 2012; Pummer et al., 2015; Dreischmeier et al., 2017) and lignin (Bogler and Borduas-Dedekind, 2020), can retain their original INA when dry heated to temperatures over 200 °C. In contrast, the assumption that mineral particles acting as INPs cannot lose any INA when subjected to heat treatment has yet to be thoroughly tested, while the question of whether a mineral reacts differently to being heated while suspended in water or while heated in air has not been addressed at all. Zolles et al. (2015) measured the change in INA of feldspars, quartz, kaolinite and Arizona Test Dust (ATD) after dry heating to 250 °C for 4-5 h and observed only minor reductions within instrumental error. No similar surveys exist for wet heating, the more commonly used form of heat treatment of samples, although a small proportion of studies that employed the wet heat test for biological INP detection included control tests with mineral suspensions including K-feldspar (O'Sullivan et al., 2014), montmorillonite (Conen et al., 2011), kaolinite (Hara et al., 2016; Hill et al., 2016) and ATD (Yadav et al., 2019). No significant changes in INA were observed in these examples except for ATD by (Yadav et al., 2019), who attributed this to the removal of an unspecified organic ice-nucleating material from the surface of the mineral.

Finally, several studies have demonstrated the apparent lability of mineral INP kept in deionised water at room temperature over hours to days, wherein the immersion mode INA gradually decreased, which has been observed with samples of K-feldspar (Harrison et al., 2016; Peckhaus et al., 2016), quartz (Harrison et al., 2019; Kumar et

al., 2019a) and ATD (Perkins et al., 2020). It is reasonable to predict that elevated temperatures could accelerate the 'ageing' behaviour seen with these minerals leading to an INA deactivation on the timescale of a biological INP heat test.

Overall, this highlights that the potential for the false positive 'detection' of biogenic INPs through the loss of mineral INA when using heat treatments has yet to be ruled out. Here, we aim to validate the heat test in its current form by fully characterising how mineral INPs respond to heating both in air and in water compared to biogenic INPs. We achieve this via a laboratory study in which we tested the immersion mode INA of a set of atmospherically relevant mineral samples before and after two types of heat treatment. We also performed equivalent tests on a set of biogenic INP analogue samples for direct comparison to the mineral INP results and as a positive control to ensure that the heat treatments would reproduce the known heat sensitivity behaviour of biogenic INPs.

We employed two methods of heat treatment: (1) direct heating of the sample in aqueous suspension (wet heating), and (2) heating the sample while in dry powder form prior to immersion in water (dry heating). This enabled us to investigate whether the deactivation behaviour of a sample depends on the medium in which it is heated, as previous studies have either involved heating samples in the wet or dry modes but not both. Our rationale for this is that an INP sample's potential chemical or physical reactions to heating in water or air may differ as these are fundamentally different treatments. Where possible, we also characterise the heat sensitivity of the important subclasses of mineral INPs and then discuss how this could affect interpretations of biogenic INP heat test results and how this can inform us in the development of a more robust protocol. While our primary objective was to empirically evaluate commonly employed heat tests, we also discuss the physical reasons for the changes in INA found in our results, which may prove useful for future studies on the fundamental mechanisms of how mineral surfaces nucleate ice. This work may also be pertinent to emerging practical applications for mineral-based ice-nucleating agents in fields such as cell cryopreservation (Daily et al., 2020; Wragg et al., 2020; Morris and Lamb, 2018).

## 2. Materials and methods

### 2.1 Sample selection

A set of atmospherically relevant ice-nucleating materials was assembled into two broad classes of "mineral" and "biogenic" for heat treatment experiments. The mineral class comprised samples of ground minerals (either purchased from vendors in a milled form or milled in-house from a bulk mineral using a planetary ball mill) and commercially available dust proxies. Details of the identity, provenance and purity of each of these are provided in Tables 2 and 3. Background information on each class of mineral and their significance as atmospheric INP is provided in section S1 of the Supplementary Information (SI). Most emphasis was placed on the feldspar and silica classes of minerals as these have previously been shown to be the most ice-active mineral classes in immersion mode freezing experiments (Atkinson et al., 2013; Harrison et al., 2019; Peckhaus et al., 2016) and therefore likely control the INA of a mineral dust assemblage of mixed mineralogy. Several of our samples have been analysed in the past using the same method and instrumentation as we employed here (Atkinson et al., 2013; Whale et al.,

2017; Harrison et al., 2016; Harrison et al., 2019) and of these only Atkinson Quartz showed a deviation (slight loss in activity) in INA since they were last tested. This indicates that the INA of the mineral samples remains largely stable while in storage. The remaining mineral samples were clay-based samples, the dust surrogates NX Illite and ATD and, finally, calcite.

### 2.1.1 Mineral sample selection rationale

We included five different samples of K-feldspar in our survey (see Fig. 2a) in order to represent the diversity of this group of minerals. These included: BCS-376 Microcline, which was studied previously (Atkinson et al., 2013) and is considered generally representative of the INA of standard K-feldspars (Harrison et al., 2016); Amazonite and TUD#3 Microcline, which are samples of microcline that show exceptionally high INA compared to typical variants of microcline and the other K-feldspar polymorphs for reasons that are still unclear (Harrison et al., 2016; Welti et al., 2019; Peckhaus et al., 2016); Eifel Sanidine, which exhibits much lower INA compared to the other samples due to a lack of features related to exsolution microtexture (Kiselev et al. 2021; Whale 2017}.

Three samples of plagioclase feldspar were included (see Fig. 3), two of which - BCS-375 Albite (Atkinson et al., 2013; Harrison et al., 2016) and TUD#2 Albite (Peckhaus et al., 2016) - are predominantly composed of the albite endmember, and Labradorite – a plagioclase that features a Ca composition between 50 % and 70 % that of anorthite. BCS-375 Abite contains quartz (4.0 %) and K-feldspar (16.7 %) impurities, while TUD#2 Albite contains at least 90 % plagioclase feldspar with the remaining 10 % of the content being unknown (based on X-ray diffraction (XRD) data (Atkinson et al., 2013; Peckhaus et al., 2016). The presence of K-feldspar in the former may mean that the observed activity is related to the presence of this component. No information is available on the mineral impurities present in the Labradorite sample. However, as plagioclase feldspar of labradorite composition is typically only found in basalts and gabbros, it is unlikely to coexist with quartz or K-feldspar since these rarely occur in these types of igneous rocks.

We included three samples of silica (see Fig. 4a): two $\alpha$-quartz samples - Atkinson Quartz and Fluka Quartz, and a sample of Fused Quartz, which is produced by melting quartz crystals then quenching to yield a glass (i.e. amorphous silica). We also included Bombay Chalcedony, which is composed of a cryptocrystalline intergrowth of quartz and moganite (another silica polymorph) and is notable as being the silica sample with the highest recorded INA (Harrison et al., 2019). We included this sample since its distinctly higher INA implies that the nature of the ice-active sites may be distinct from those of $\alpha$-quartz. Literature XRD data for our quartz samples indicate they are exceptionally pure in terms of mineralogy, with an $\alpha$-quartz content of at least 99.9 % (Harrison et al., 2019). Fused Quartz, as stated by the manufacturer, has a silica content of >99 %.

To represent clays we included samples that represent the main classes of clay minerals but also different samples of the same mineral to account for impurities which, due to the generally low INA of clays, may control the INA of the sample. Two samples of kaolinite (KGa-1b Kaolinite and Fluka Kaolinite), two samples of montmorillonite (SWy-2 Montmorillonite and Sigma Montmorillonite) and one sample of chlorite (Chlorite). An illite-rich sample

(NX Illite) was included, but we classed this along with Arizona Test Dust (ATD) as a Mineral Dust Analogue – these being commercially available dusts of composite mineralogy which have been used in the past as representative e surrogates of atmospheric mineral dust. Finally, powder from a ground pure calcite crystal was used to represent the carbonates.

220

**Table 2:** Sample information for mineral-based INP samples. Sources of data for purity and specific surface area (SSA) are detailed in the annotations.

| Sample name | Classification | Source | Purity (%) | SSA ($m^2 g^{-1}$) |
|---|---|---|---|---|
| BCS-376 Microcline | K-feldspar | Bureau of Analysed Samples, UK | 80.1[a] | 2.59 |
| TUD#3 Microcline | K-feldspar | TU Darmstadt, Germany | 80.0[b] | 2.94[b] |
| Amazonite Microcline | K-feldspar | University of Leeds mineral collection | No data | No data |
| Eifel Sanidine | K-feldspar | University of Leeds mineral collection | No data | 1.1[c] |
| Pakistan Orthoclase | K-feldspar | University of Leeds mineral collection | No data | No data |
| TUD#2 Albite | Plagioclase feldspar | TU Darmstadt, Germany | 90[b] | 1.92[b] |
| BCS-375 Albite | Plagioclase feldspar | Bureau of Analysed Samples, UK | 76.6[a] | 5.8[a] |
| Atkinson Quartz | Quartz | University of Leeds mineral collection | 99.9[d] | 4.2[d] |
| Fluka Quartz | Quartz | Honeywell/Fluka (Cat No 83340), UK | 99[d] | 0.9[d] |
| Fused Quartz | Quartz | Goodfellow (Cat No. SI616010, 45 µm), UK | >99[*] | No data |
| Bombay Chalcedony | Quartz | University of Leeds mineral collection | >99[d] | 1.23[d] |
| KGa-1b Kaolinite | Clay based | Clay Minerals Society, USA (KGa-1b) | 96[a] | 13.6[a] |
| Fluka Kaolinite | Clay based | Fluka (Cat No. 03584) | 82.7[a] | No data |
| Sigma Montmorillonite | Clay based | Sigma Aldrich (Cat No. 69907), UK | 57.0[a] | No data |
| SWy-2 Montmorillonite | Clay based | Clay Minerals Society, USA | 75[a] | 91.4[a] |
| Chlorite | Clay based | University of Leeds mineral collection | 99.6[a] | 25[a] |
| Arizona Test Dust (ATD) | Dust surrogate | Powder Technology Inc., USA (A1 Ultra fine) | - | 85[e] |
| NX Illite | Dust surrogate | Arginotec, B+M Nottenkamper, Germany | - | 104.2[f] |
| Calcite | Carbonate | University of Leeds mineral collection | 99.6[a] | 6[a] |

References: a. Atkinson et al. (2013), b. Peckhaus et al. (2016), c. Whale et al. (2017), d. Harrison et al. (2019), e. Bedjanian et al. (2013), f. Broadley et al. (2012), * Data from manufacturer

225

### 2.1.2 Biogenic sample selection rationale

The biogenic class of samples tested here included examples of material with expected INA heat sensitivity in which proteins are responsible for ice nucleation (Snomax® as a non-viable form of Pseudomonas syringae bacteria, and lichen collected from trees in southern Finland). We also tested a non-proteinaceous material (microcrystalline cellulose (MCC) powder as well as and silver birch pollen that has been shown to have some heat resistance. The sources for each sample are provided in Table 3. Snomax® and lichen were used as representative of bacterial and fungal derived proteinaceous INP respectively. Snomax® is a snow inducer product composed of lyophilised material derived from *Pseudomonas syringae* bacteria cultures and also used a surrogate for ice-nucleating bacteria (Wex et al., 2015; Polen et al., 2016). Lichens are symbiotic associations of fungi and algae and have been found to contain highly active INPs that are proteinaceous and likely originate from the fungal component (Moffett et al., 2015; Kieft and Ruscetti, 1990) so was therefore used a convenient source of fungal ice-nucleating material. Fungal INPs, have been found to have slightly higher heat resistance in wet mode

compared to bacterial INPs, typically showing no reduction in INA with up to 60 °C of heating compared to 40 °C with bacterial INP but for both these it is eliminated by heating above 90 ° C (Pummer et al., 2015; Pouleur et al., 1992; Fröhlich-Nowoisky et al., 2015). Pollen-based INP were chosen to represent a heat resistant INP type while little is known about the heat resistance of cellulose (Pummer et al., 2012; Bogler et al., 2020). A sample of raw silver birch pollen (*Betula pendula*) was used to prepare an aqueous suspension of birch pollen washing water (BPWW). Pollen contains water-soluble macromolecules (Pummer et al., 2012; Dreischmeier et al., 2017) which can be readily released into suspension with water (Augustin et al., 2013). Their ice-nucleating activity has been linked to polysaccaride components (Dreischmeier et al., 2017; Pummer et al., 2012) partly owing to their relative heat resistance (Pummer et al., 2012), although other lines of evidence point to the involvement of a proteinaceous component (Burkart et al., 2021; Tong et al., 2015). Microcrystalline cellulose (MCC), a particulate polysaccharide reagent derived from wood pulp, was used as a surrogate for detrital plant material (Hiranuma et al., 2015b).

**Table 3:** Sample information for biological-based INP samples.

| Sample name | Classification | Source | SSA (m$^2$ g$^{-1}$) |
|---|---|---|---|
| Snomax® (freeze-dried, non-viable *Pseudomonas syringae* bacteria) | Proteinaceous | York Snow, Inc., USA | Non-particulate |
| Lichen (*Evernia prunastri*) | Proteinaceous | Collected from trees around Hyytiälä Forestry Station, Finland | Non-particulate after 0.2 μm filtration |
| Birch pollen (*Betula pendula*) washing water (BPWW) | Proteinaceous or polysaccharide (see text) | Pharmallerga, Czech Republic | Non-particulate after 0.2 μm filtration |
| Microcrystalline cellulose (MCC) | Polysaccharide | Sigma-Aldrich, UK (Cat. No. 435236) | 0.068[a] |

**2.2 Sample preparation**

All aqueous suspensions were prepared in 0.1 μm pre-filtered, cell culture-grade deionised water (HyClone™ HyPure, GE Lifesciences). A standard concentration of 1 % w/v was used for the mineral suspensions although additional aliquots with different concentrations were prepared for selected mineral samples (BCS-376 Microcline, Fluka Quartz, NX-Illite and ATD – see Appendix B). All mineral samples were stored in darkness at room temperature and suspensions of mineral samples were prepared by mixing 0.1 g of sample with 10 mL water in 20 mL borosilicate glass vials (Samco type T006/01, Surrey, UK), which had been pre-sterilised by dry heating at 175 °C for 2 h. The suspensions were thoroughly dispersed on a vortexer for 10 seconds after mixing and were gently inverted by hand several times to ensure even mixing before each drawing of the suspension via an electronic pipette. Glass vessels, rather than polypropylene centrifuge tubes, were used for all experiments unless stated due to their higher thermal conductivity when placed in a water bath (see Appendix A) and resistance to oven heating. The biogenic samples were stored cold (Snomax® at −20 ˚C, raw birch pollen and dried lichen at −4 ˚C) or at room temperature in the case of MCC and they were made to suitable concentrations according to existing literature protocols. Snomax® and MCC were prepared as 0.05 % w/v and 0.1% w/v suspensions respectively in the same manner as the mineral samples. Lower concentrations of Snomax® were made by dilution of the 0.05% w/v suspension with deionised water. The birch pollen and lichen samples could not be immediately dispersed in water as they required additional filtration steps to produce visibly clear homogeneous extracts rather

than particulate suspensions. A suspension of birch pollen washing water (BPWW) was prepared similar to a previously described protocol (O′Sullivan et al., 2015), wherein a 2 % w/v (20 mg mL$^{-1}$) suspension of raw pollen powder was prepared by weighing 0.2 g of raw pollen and adding to 10 mL deionised water, before mixing via vortexing and shaking, and being allowed to soak overnight at 4 °C. Previous iterations of the filtration protocol involved filtering through an 11 µm nylon net filter prior to a 0.2 µm cellulose acetate filter in order to pre-filter out larger sample debris. This stage was, however, omitted as handling blanks suggested that the stainless steel housing and associated PTFE O rings for the nylon net filter would leach out INP after being used to filter raw birch pollen (see Fig. S3) despite cleaning. Instead, the raw pollen suspension was then filtered directly through a disposable 0.2 µm cellulose acetate filter which had been pre-flushed with deionised water as it was found this method did not produce a significant handling blank signal (Fig S3). The resulting clear filtrate left only the washing water containing macromolecular INPs for analysis. The lichen sample (*Evernia prunastri*) was collected from a tree during a field campaign at the Hyytiälä Forestry Station, Finland, in April 2018, was air dried and then stored in a sealed sterile plastic bag at 4 °C. The dried lichen was cut into millimetre-sized pieces, then 5 mL of purified water added to yield a lichen concentration of 2 % w/v and placed on a rotary inverter for 2 h on a slow setting. The suspension was then filtered as per the BPWW procedure above to produce an aliquot of extract ready for analysis with or without subsequent dilution.

## 2.3 Heat treatments

Each INP sample was subjected to heat treatment using two distinct methods: a 'wet heating' treatment wherein the INP was heated while in suspension and a 'dry heating' treatment wherein the INP sample was heated in dry powder form in air and subsequently mixed with water for analysis. The 'standard' temperature and duration for the wet heat test was 95 °C for 30 min (immersed in boiling water – see Figure A2) and for the dry heat test was 250 °C for 4 h. These conditions have been used in in previous work and our primary objective was to empirically test commonly used heat tests. For selected samples we varied heating temperature (for the dry test only) and durations for further analysis of the samples' responses.

### 2.3.1 Wet heating

The 'wet heating' treatment comprised of a sealed vessel containing the INP suspensions, described in Section 2.2, being immersed in an open boiling water bath (hence at the boiling point of water at 1 atm: 100 °C). The vessels were sealed tightly to prevent the evaporation of water from the vessel causing an increase in concentration of the suspensions. After 30 min of immersion, the vessel was removed from the water bath and then allowed to cool to room temperature prior to the droplet freezing assay. In the case of the washing water samples (i.e., BPWW and lichen), the wet heat treatments took place following the filtration steps.

The temperature profile of the liquid inside 20 mL borosilicate glass vials and 50 mL polypropylene centrifuge tubes (Corning Falcon 352090) throughout the wet heat treatment procedure was measured (Fig. A2 in Appendix A) by inserting a thermocouple (Type K) through a small hole punched in the caps of the vessels, and was recorded using a data logger (TC-08, ±0.025 °C, Pico Technology, UK). This showed that a 10 mL liquid aliquot inside the glass vial reached a maximum temperature of 96 °C after approximately 10 min of immersion in the boiling water bath, while the larger polypropylene vessel only reached 86 °C after 20 min. Samples of proteinaceous IN derived

from lichen (Kieft 1988, Kieft and Ruscetti 1990, Moffat et al, 2015), fusarium fungi (Poleur et al. 1992) and psuedomonas syringae bacteria (Maki et al. 1974) all saw large deactivations after being heated in boiling water baths for less than 15 minutes, presumably due to denaturation. Therefore, it is presumed that 30 min of immersion under these conditions is sufficient to denature INP of proteinaceous origin. Both vessels returned to ambient temperature approximately 45 min after being removed from the bath.

In addition to our standard 30 min heat treatment, we performed extended wet heat treatments of up to 24 h for selected samples by immersing the vessels in a silicone oil bath heated to 100 °C rather than a water bath. The oil bath temperature was controlled using a thermostat alongside a magnetic stirrer bar and stirrer plate to ensure a homogeneous oil temperature.

### 2.3.2 Dry heating

'Dry heating' of samples was achieved by placing a 20 mL borosilicate glass vial containing a maximum of 0.2 g of dry sample in a standard laboratory oven at 250 °C for 4 h, before being allowed to cool to room temperature and then prepared as an aqueous suspension as described in Section 2.2 above. The dry heat treatment for the biogenic samples were performed on the raw, dry materials, then suspended in deionised water and, in the case of the BPWW and lichen, subject to the filtration process described above. A temperature of 250 °C for 4 h was selected for the 'standard' dry heat treatment as it far exceeded the highest documented temperature at which the most heat resistant biogenic INPs (birch pollen at around 180 °C (Pummer et al., 2012) and lignin at around 220 °C (Bogler et al, 2020)) are deactivated. Further, this temperature is lower than the maximum heat rating of polytetrafluoroethylene (PTFE) membrane and quartz filters that are often used to collect aerosol samples for INP analysis. We also Samples were weighed before and after standard dry heat treatment.

### 2.4 Ice nucleation measurements by droplet freezing assay and determination of samples' heat deactivations

The INA of the mineral-based and biogenic sample suspensions both before and after heat treatments was determined by performing immersion mode droplet freezing assays (Vali, 1971) using the Microlitre Nucleation by Immersed Particle Instrument (μL-NIPI) (Whale et al., 2015), which has been used extensively for INP analysis in the literature and in several intercomparison studies (Hiranuma et al., 2015a; DeMott et al., 2018). Here, 1 μL droplets (up to a maximum of 50) of INP suspension were pipetted onto a hydrophobic glass coverslip (22 mm diameter, cat. no. HR3-231, Hampton Research, USA) that was located atop the aluminium cooling plate of a Grant-Asymptote EF600 cryocooler whilst at room temperature. The cooling plate was then enclosed in a Perspex chamber into which a flow of dry nitrogen gas was introduced at 0.3 L min$^{-1}$ to flush the chamber of ambient air and to prevent the presence of moisture and airborne contaminants for the duration of each experimental run. The droplets were cooled at a rate of 1 °C min$^{-1}$ until all of the droplets were frozen. Droplet freezing events were detected visually using an optical camera (Microsoft LifeCam HD) mounted atop the clear Perspex flow chamber. Analysis of the droplet freezing events allowed the determination of the fraction of droplets frozen as a function of temperature, $f_{ice}(T)$, as shown in Eq. (1), where $n(T)$ is the number of droplets frozen at temperature $T$, and $N$ is the total number of droplets in the assay.

$$f_{ice}(T) = \frac{n(T)}{N}$$ (1)

Blank tests were performed with droplets of filtered deionised water at the beginning of each day of experiments to confirm no contamination was introduced by the processes (see Figs. S1 and S2). In addition we performed handling blanks for the various heating methods and for the filtration of the BPWW and lichen samples (Fig. S3). Quantification of a nucleator's INA was achieved by determining the surface density of ice-active sites, $n_s(T)$ for particulate samples (minerals and MCC), or mass density of active sites, $n_m(T)$, for non-particulate biogenic samples. These allowed for comparison of our data with existing parameterisations for ice-nucleating materials (e.g., Fig 1b for BCS-376 Microcline or Fig B1e for Snomax®): If the surface area of nucleant present in each droplet, $A$, is known, then this can be used to calculate $n_s(T)$ of an INP sample from $f_{ice}(T)$, as defined in Eq. (1).

$$n_s(T) = \frac{-\ln(1 - f_{ice}(T))}{A} \tag{2}$$

Similarly if the mass of nucleant present in each droplet, $M$, is known, $n_m(T)$ can be calculated:

$$n_m(T) = \frac{-\ln(1 - f_{ice}(T))}{M} \tag{3}$$

Mass per droplet $M$ is calculated from droplet size and the concentration of the INP suspension while the surface area, $A$, in our samples was derived mass of ice nucleant per droplet multiplied by the specific surface area (SSA, $m^2\,g^{-1}$) of the INP powders. We used literature values of SSA obtained by the Brauner-Emmet-Teller (BET) $N_2$ gas adsorption technique (e.g. Harrison et al., 2019; Zolles et al., 2015; Paramonov et al., 2018) for all mineral samples apart from SSA of the BCS-376 microcline K-feldspar which we measured ourselves (Micrometrics TriStar 3000). For MCC we used a scanning electron microscope (SEM) based measurement from Hiranuma et al. (2019). This INA quantification approach assumes that each ice nucleation site has a characteristic temperature at which it always becomes active and time dependence is insignificant (Herbert et al., 2014), otherwise known as the singular description of heterogeneous ice nucleation (Koop and Zobrist, 2009; Murray et al., 2012; Vali, 1994; Pruppacher, 1978). Characterising ice-nucleating materials in terms of $n_s(T)$ also forms the basis of models used for predicting the temperatures (and thus cloud regime) at which different classes of atmospheric INPs may become active (Vergara-Temprado et al., 2017; Hawker et al., 2021; Zhao et al., 2021; Murray et al., 2012).

## 3. Results and discussion

In Fig. 1 we have shown several examples of fraction frozen curves for heated (wet and dry) and unheated samples to illustrate the heat sensitivity of a range of ice nucleating materials. Similar plots for all materials we have tested here are shown in Figs. S1 and S2. In order to present this information in a more concise manner, we have plotted the same data in the form of boxplots of droplet freezing temperatures of mineral samples throughout the results section. In addition, changes in INA resulting from the heat treatments were evaluated by calculating the freezing temperature shifts between them and determining whether the shifts were significantly larger than the instrumental error. This was simply taken as the difference between the median droplet freezing temperature ($T_{50}$) of the samples before and after the heat treatments to give a $\Delta T_{50}$ value: $\Delta T_{50}^{wet}$ for the wet heat treatment and $\Delta T_{50}^{dry}$ for the dry heat treatment. Droplet freezing temperatures detected by the µL-NIPI instrument have a nominal error of ±0.4 °C (Whale et al., 2015) so, as a simple test for significance, a change in $\Delta T_{50}$ by more than 3 times this ±0.4 °C error (i.e., ±1.2°C) between $f_{ice}(T)$ curves qualified as a significant shift. A significant shift in $T_{50}$ to colder temperatures (i.e., a negative $\Delta T_{50}$ value) indicated a deactivation in INA of a sample in response to heating.

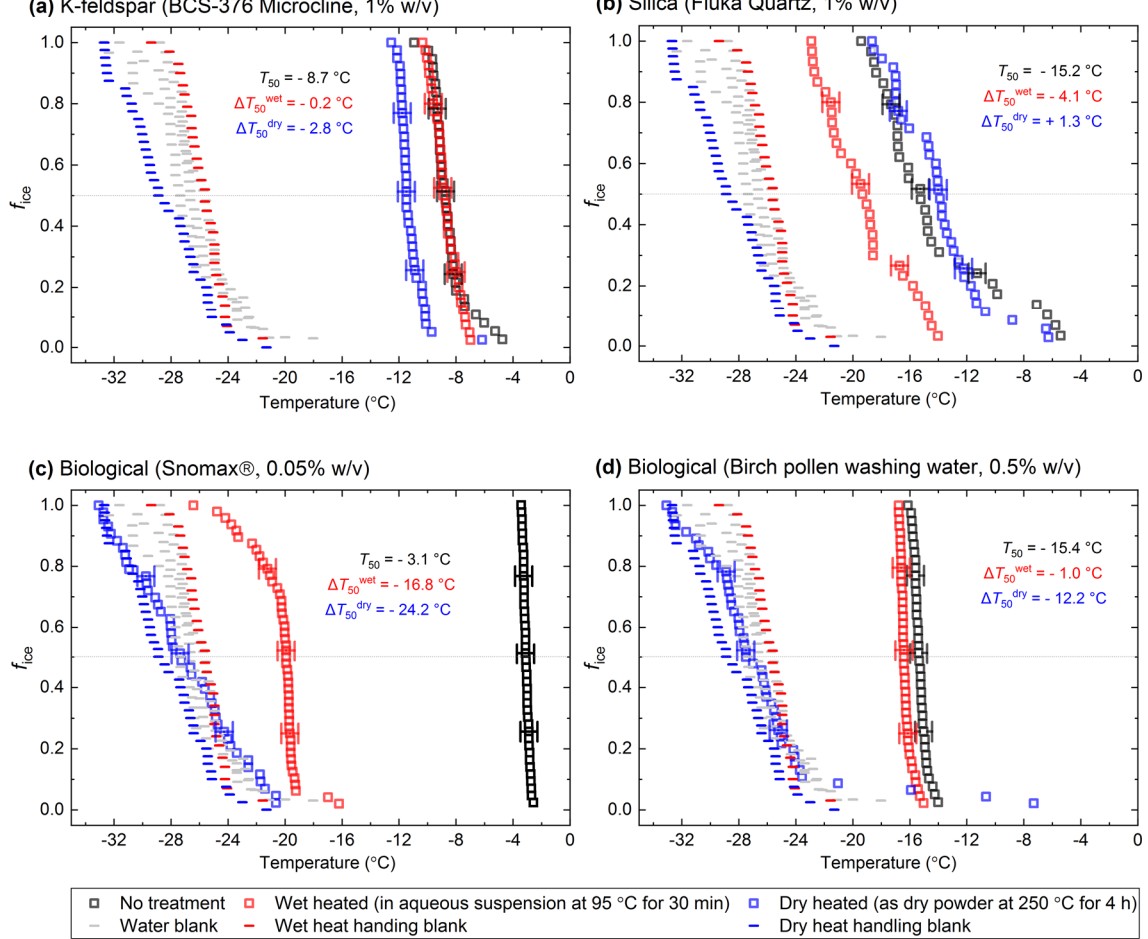

**Figure 1.** Fraction of droplets frozen ($f_{ice}(T)$) spectra illustrating characteristic heat treatment responses for: **(a)** A dry heat-sensitive mineral INP (BCS-376 Microcline). **(b)** A wet heat-sensitive mineral INP (Fluka Quartz). **(c)** A wet heat-sensitive biological INP (Snomax®). **(d)** A wet heat-insensitive biological INP (birch pollen washing water). Clean water blanks and handling blank $f_{ice}(T)$ curves are shown as well as a dotted line is shown at $f_{ice}(T) = 0.5$, at which $T_{50}$ temperatures can be read. Error bars on selected data points illustrate the 1.2 °C temperature range used to guide whether a shift in freezing temperatures after heating was considered to be significant.

## 3.1 Heat sensitivity of mineral-based INP samples

### 3.1.1 K-feldspars

In general, the INA of K-feldspar samples did not respond substantially to wet heating for 30 mins with no significant reductions of $T_{50}$ in four out of five of the samples of K-feldspar (see Fig. 1a and 2). An exception was Amazonite Microcline which showed a $\Delta T_{50}^{wet}$ of −1.5 °C, which was greater than the experimental uncertainty, as defined above (we discuss the possible reasons for this later in this section). The finding that K-feldspars are relatively insensitive to the wet heat test is consistent with the findings of O'Sullivan et al. (2014) and Peckhaus et al. (2016), who previously performed this test on BCS-376 Microcline and TUD#3 Microcline, respectively. Additional wet heat tests on less concentrated (0.1% and 0.02%) suspensions of BCS-376 Microcline also showed no response (Appendix B) indicating stability over a wide range of particle concentrations. We also conducted extended wet heat treatments of up to 22 h with BCS-376 Microcline that, although longer in duration than typical

biological INP heat tests, were designed to ascertain whether wet deactivation was possible. The results, shown

in Fig. 2b, are plotted in the form of $n_s(T)$ to enable comparison with literature data. The results show that 24 h

of wet heating resulted in a reduction in $n_s(T)$ of approximately one order of magnitude, or 2 °C at $n_s(T) = 1$ cm$^{-2}$.

However, some deactivation of the most active sites appeared to occur after only 60 min of heating. After 4 h, the

reduction in $n_s(T)$ was roughly the same as that seen after 16 months of immersion in water at room temperature,

as determined by Harrison et al., (2016). Overall, we find that the INA of K-feldspar is retained after short term

(30 min) wet heating but can be reduced if heated for longer periods.

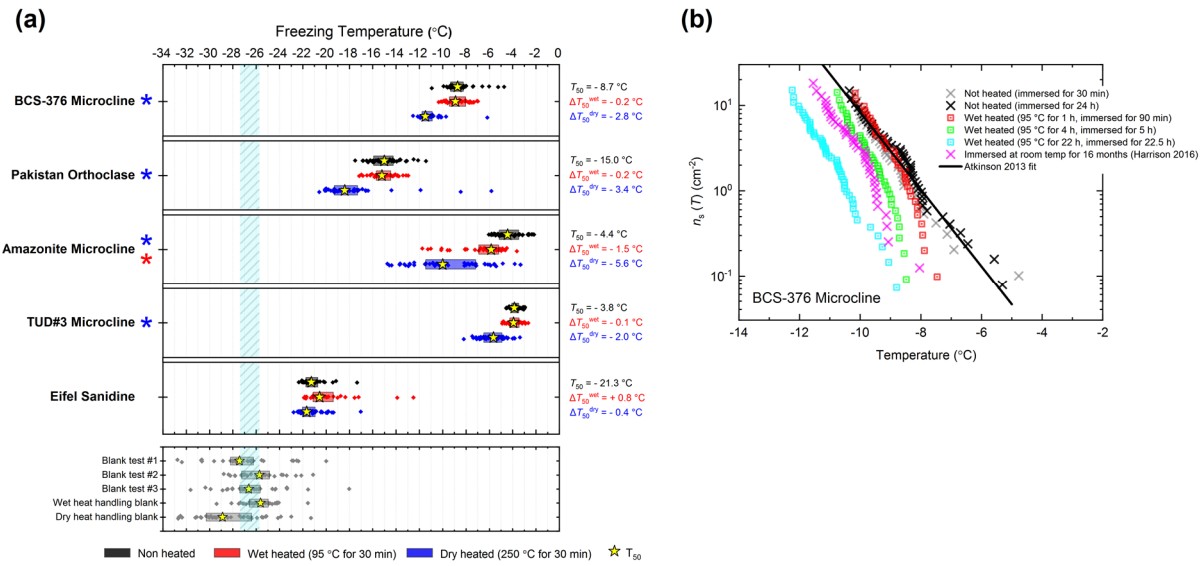

**Figure 2. (a)** Boxplot showing freezing temperatures before (black) and after heat treatments (red for wet heat, blue for dry heat) for all K-feldspar samples. Asterisks (*) next to sample name indicate significant deactivation by wet (red) or dry (blue) heating, this defined as a reduction in $T_{50}$ of more than 1.2 °C. Boxes represent the 25-75 % percentile, points represent individual droplet freezing temperatures, and stars represent the temperature at which half of the droplet population had frozen (i.e. $T_{50}$). Clean water blank freezing temperatures are illustrated by the blue band, which denotes the range of $T_{50}$ temperatures obtained from four blank droplet freezing runs. Blank runs and handling blank runs are shown below the main boxplot **(b)** Active site density per surface area ($n_s(T)$) spectrum for BCS-376 Microcline after extended wet heat treatment compared to room temperature ageing experiments from Harrison et al. (2016). The parameterisation for the ice-nucleating activity of K-feldspar from Atkinson et al. (2013) is also shown.

To discuss the reasons behind the deactivation of K-feldspar when wet heated for longer than 30 min, the nature

of the ice-nucleating sites on minerals must first be considered. Ice nucleation on mineral surfaces such as

feldspars has been shown to occur at specific sites that become active at a specific temperature (Holden et al.,

2019; Holden et al., 2021). Topographical features associated with exsolution microtexture (Whale et al., 2017;

Kiselev et al., 2021) have been proposed as the locations of the highly active sites on K-feldspar. Moreover,

Kiselev et al. (2017) observed that ice crystals growing from the vapour phase on the surface of microcline

originated on steps and cracks and were preferentially orientated between the basal face of ice and the (100)

cleavage plane. More recent work suggests that cracks caused by exsolution microtexture may expose the (100)

face of feldspars (Kiselev et al., 2021). The chemical and physical nature of these sites is still unclear, however

molecular dynamics studies such as those by Pedevilla et al. (2017) show that having a high density of functional

groups like silanol groups (Si-OH), where water can hydrogen bond with the mineral surface and potentially order

(such as those exposed at the (100) cleavage plane), may be important for nucleating ice (Harrison et al., 2019).

The most obvious physical cause of the INA deactivation of K-feldspar by wet heating would be the alteration of the mineral surface by dissolution via hydrolysis. This leaves an amorphous 'leached' layer at the surface (Lee et al., 2008; Chardon et al., 2006), destroying or at least disrupting the ice-active sites. Several studies have shown experimentally that acid treatment deactivates K-feldspar INPs (Augustin-Bauditz et al., 2014; Kulkarni et al., 2015; Kumar et al., 2018a). In pure water and at near-neutral pH, however, the supply of $H^+$ for hydrolysis is lower and therefore the dissolution rate is much slower, but may still occur at the less stable, higher energy sites and topographic features (Parsons et al., 2015), which are themselves proposed as the highly active sites in K-feldspars. As discussed above, Harrison et al. (2016) observed a gradual INA deactivation of BCS-376 Microcline while at room temperature in deionised water, but this occurred over several months rather than hours. It is reasonable to propose that the same INA deactivation process observed by Harrison et al. (2016) also occurred on the K-feldspar samples in this study but was accelerated in this case by heating.

Amazonite Microcline, one of our two highly ice-active microcline samples, was an exception to other K-feldspar samples in that short-term wet heating resulted in a significant but small deactivation ($\Delta T_{50}^{wet} = -1.5$ °C). This could be because either the highly active sites of this sample were especially susceptible to dissolution and distinct from the more standard sites in the other K-feldspar samples, or is an indication of contamination with biological INPs. We return to this issue below.

Dry heating had a stronger deactivating effect on the K-feldspar samples than wet heating (Fig. 2a). Amazonite Microcline showed the largest $\Delta T_{50}^{dry}$ of $-5.6$ °C and we observed that this sample lost its pale green colour and turned white following the treatment, becoming more similar in appearance to the other K-feldspar samples. Dry heating resulted in deactivation of the Pakistan Orthoclase ($\Delta T_{50}^{dry}$ of $-3.4$ °C) and produced smaller deactivations in TUD#3 Microcline and BCS-376 Microcline ($\Delta T_{50}^{dry}$ of $-1.8$ °C and $-2.8$ °C, respectively).

A potential alternative explanation for the apparent dry-heat sensitivity of K-feldspar is that there is a biological component mixed with the K-feldspar samples which nucleates ice and is deactivated on heating and, due to the high temperatures required for deactivation, is unlikely to be bacteria- or fungal-derived. Peckhaus et al. (2016) discussed the potential for biological ice nucleating material in TUD#3. They achieved a deactivation in TUD#3 Microcline by treatment with hot aqueous $H_2O_2$ after a wet heat test (90 °C for 1 hr) showed no effect. They deliberated the presence of an polysaccharide based ice-nucleating component, but concluded this was unlikely given the unrealistic mass proportion of contaminant in the sample that would be required to produce such high ice-nucleating temperatures to start with. In the case of our affected K-feldspar samples, presence of heat-resistant biogenic material cannot be completely ruled out without further analysis, however there are no likely candidates this kind of material that nucleate ice at such high ( > -5 °C) temperatures. Hence, we suggest that the deactivation observed on dry heating K-feldspars is not related to the destruction of biological material.

Recalcitrant organic coatings have previously been proposed as the source of INA in mineral dusts that is lost upon dry heating (Paramonov et al., 2018; Peckhaus et al., 2016; Perkins et al., 2020). however others have reported that organic coatings suppress the INA of mineral dusts rather than enhance it (Boose et al., 2019; China

et al., 2017; Pach et al., 2021) by blocking access to underlying active sites. For example, Pach et al (2021) treated slices of a K-feldspar crystal from the same locality as TUD#3 Microcline with oxygen plasma and observed an enhancement in INA which they attributed to the oxidation and removal of organic material from the surface that originated from ambient air. They suggetsed that the plasma treatment 'unblocked' the surface pores which contained the most active IN sites allowing water to enter during their freezing experiments. .

Alternatively, the loss of a (non-organic) volatile component during dry heating may alter K-feldspar in a way that reduces its INA. As described above, Amazonite Microcline is a green or turquoise coloured variant of microcline and was observed here to lose its green colouration upon dry heating. This phenomenon has previously been observed (Hofmeister and Rossman, 1985) and was correlated to the loss of water molecules that were structurally bound within the feldspar crystal lattice. Although Amazonite is a relatively rare variety of microcline, all feldspars contain a minor water component either as lattice-bound $H_2O$ molecules or OH groups or fluid pockets (Johnson and Rossman, 2003) that can be driven off by high temperature (Liu et al., 2018), as could be the case in our dry heat treatment. Although it is not known whether this process would destroy the active ice nucleation sites, it is intriguing that microcline samples, the most ice-active variety, when surveyed were found to contain the most structurally bound water of all feldspars (Johnson and Rossman, 2004).

### 3.1.2 Plagioclase feldspars

BCS-375 Albite and TUD#2 Albite showed no significant changes to their $T_{50}$ values after wet heating, but both samples lost much of the tail of INA that extended to above −15 °C (Fig. 3 and Figs. S1f and S1g). Labradorite was sensitive to wet heating, with a $\Delta T_{50}^{wet}$ of −2.4 °C. All three samples in this mineral class were found to be insensitive to dry heat treatment, which means that biological contaminants was unlikely to be the source of the wet heat-labile INA (since all biological samples we looked at were sensitive to dry heat, see Section 3.2). The wet heat sensitivity was consistent with accelerated dissolution of the mineral surface, as discussed above for K-feldspars (and below for quartz). The Si dissolution rate for plagioclase feldspar is similar to that of quartz, but two orders of magnitude higher than that of K-feldspar microcline (Kumar et al., 2019a). This was consistent with the observation that, for example, both Labradorite and Fluka Quartz (see Sections 3.1.2 and 3.1.3) deactivated after 30 min of wet heating while BCS-376 Microcline took several hours to show deactivation. The wet heat reactions of the two albite samples were more difficult to interpret due to their varying impurities of other feldspars and quartz, and as such the deactivations could be a result of deactivation of those impurities rather than the plagioclase feldspar itself.

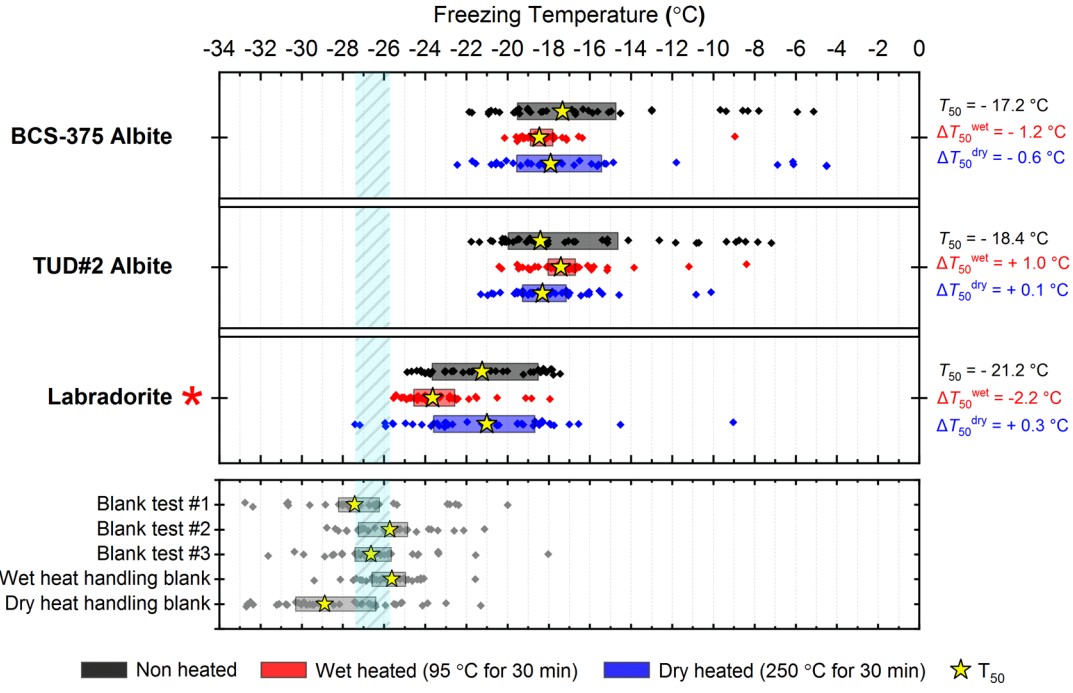

**Figure 3. (a)** Boxplot showing freezing temperatures before (black) and after heat treatments (red for wet heat, blue for dry heat) for all plagioclase feldspar samples along with clean water blank and handling blank runs.

### 3.1.3 Silicas

Atkinson Quartz, Fluka Quartz and Fused Quartz all exhibited similar reactions to both wet and dry heat treatments (see Fig. 4a). In each case, the INA experienced significant deactivation upon wet heating ($\Delta T_{50}^{wet}$ of −7.3, −4.1 and −4.4 °C for Atkinson Quartz, Fluka Quartz and Fused Quartz, respectively), but no significant changes when dry heated. We repeated the standard wet and dry heat tests on Fluka Quartz at higher (2.5%) and lower (0.1%) suspension concentrations (Appendix B) and saw that the wet heat deactivation was consistent at approximately one order of magnitude of $n_s(T)$ and dry heat consistently non-deactivating, apart from a small number of droplets active above –6 ˚C. In contrast, Bombay Chalcedony showed no significant change in INA following either type of heat treatment. Kumar et al. (2019a) proposed that in their study that the quartz INP suspensions deactivated only as an artefact of being contained in glass vessels, however when we repeated our wet heat treatment in plastic containers deactivation was still observed (see Figs. A1 and A2 and Appendix A for discussion). While the INA deactivation of quartz by dry heating has previously been described by Zolles et al. (2015), this is the first time that the wet heat treatment and resultant INA lability of quartz has been reported.

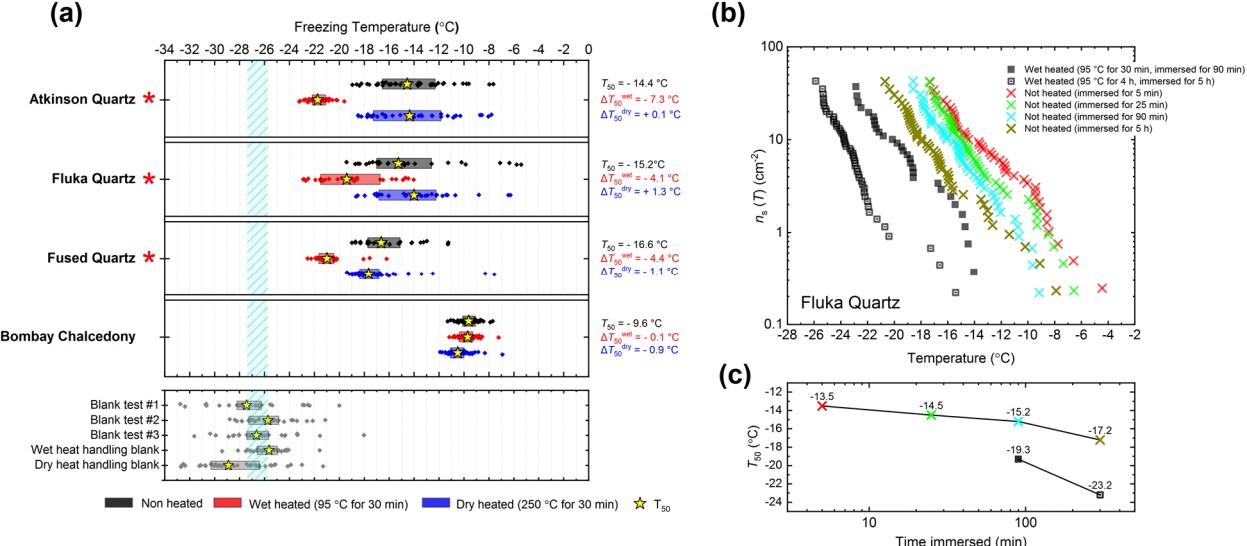

**Figure 4. (a)** Boxplot showing freezing temperatures before (black) and after heat treatments (red for wet heat, blue for dry heat) for all silica samples along with clean water blank and handling blank runs. **(b)** $n_s(T)$ spectrum for Fluka Quartz after wet heat treatment and room temperature ageing, illustrating their relative rates of INA deactivation. **(c)** Time series of $T_{50}$ values for data plotted in (b).

Being sensitive to wet heat, yet virtually resistant to dry heat treatment, is an indirect but strong indication that the heat labile ice-nucleating sites on Atkinson Quartz, Fluka Quartz and Fused Quartz are not biological in nature. This is because our dry heat treatment would be expected to reduce the activity all biological INPs (as our results with biogenic materials show in Section 3.2). In addition, it is interesting that the glassy Fused Quartz sample had very similar responses to both dry and wet heat. This indicates that the active sites on these three silica samples are not dependent on crystallinity. Given that Bombay Chalcedony was the exception in this mineral class in that it was insensitive to heat, it seems that the active sites on this material were distinct to the other silica samples we studied. The high INA and stability to heat of Bombay Chalcedony is comparable to several of the K-feldspar samples. Bombay Chalcedony is also a microcrystalline material possessing micropores, much like K-feldspar, and this may give rise to stable active sites (Harrison et al., 2019).

Given the INA deactivation in silica samples upon heating appears to be abiotic and only occurs in water, but not dry heat, then the most obvious explanation is that it is due to the accelerated dissolution of surface features associated with the active sites. Active sites are thought to be most abundant where defects and fractures occur, as milling has consistently been found to increase the INA of quartz (Zolles et al., 2015; Kumar et al., 2018b; Harrison et al., 2019). They may also be the most unstable sites as Harrison et al. (2019) observed measurable 'ageing' in quartz samples (including Atkinson Quartz and Fluka Quartz) that were immersed in room temperature water for only 1 h. Our wet heat treatment of Atkinson Quartz resulted in an INA deactivation of similar magnitude ($\Delta T_{50}^{wet}$ around 7 °C) to that achieved after 16 months of aqueous room temperature ageing by Harrison et al. (2019). To demonstrate the 'accelerated' deactivation speeds of quartz in water at different temperatures, we performed parallel room-temperature 'ageing' and also wet heating (30 min and 4 h) experiments with Fluka Quartz, with the results shown in Fig. 4b. When the heated sample and room-temperature sample had both been immersed in water for the same amount of time, the heated sample always had a lower activity. In addition, the

longer the sample was immersed in water (heated or room temperature), the greater the deactivation (where the deactivation was accelerated at higher temperatures).

A similar apparent phenomenon of room-temperature ageing being accelerated by heating has also been observed for BCS-376 Microcline K-feldspar (Harrison et al., 2016), except that the process appears to be much slower. At

room temperature, INA deactivations of similar magnitude (up to 2 °C) were observed after only 1 h for Atkinson Quartz (Harrison et al., 2019), compared to 16 months required for deactivation of BCS-376 Microcline (Harrison et al., 2016). Similarly, we needed to wet heat K-feldspar for at least 1 h to detect a small deactivation compared to 30 min for Fluka Quartz. However, crucially the deactivation of quartz is, unlike K-feldspar, fast enough to occur on a timescale relevant to biogenic INP heat tests (about 30 min). It is possible that the same deactivation

mechanism for both K-feldspar and Atkinson Quartz occurs during the wet heat treatments and is consistent with active site degradation by surface dissolution for two possible reasons. Firstly, surface dissolution rates for quartz are faster than for microcline ($10^{-13}$ to $10^{-12}$ Si-m$^{-2}$ s$^{-1}$ compared with 4 x $10^{-14}$ to 8 x $10^{-14}$ Si-m$^{-2}$ s$^{-1}$ at neutral pH and 25 °C (Kumar et al., 2019b)). Secondly, quartz and feldspar break apart differently when ground. Quartz, lacking cleavage planes, fractures conchoidally, while feldspar can more easily cleave along its two perfect

cleavages situated on the (001) and (010) faces. Fracturing rather than cleaving may result in a surface topography dense in high energy but unstable ice nucleation sites that are more susceptible to dissolution (Harrison et al., 2019) than the bulk of the surrounding surface.

### 3.1.4 Clay-based mineral samples

While neither kaolinite sample were significantly sensitive to dry heat (the Fluka sample was only marginally sensitive), the Fluka Kaolinite showed clear sensitivity to wet heating while KGa-1b Kaolinite did not (see Fig. 5). We can perhaps attribute this to the comparatively purer state of the latter (96 % kaolinite) compared to the former (83 %) that includes a 6 % component of quartz, which was shown to be sensitive to wet heating in Section 3.1.3.

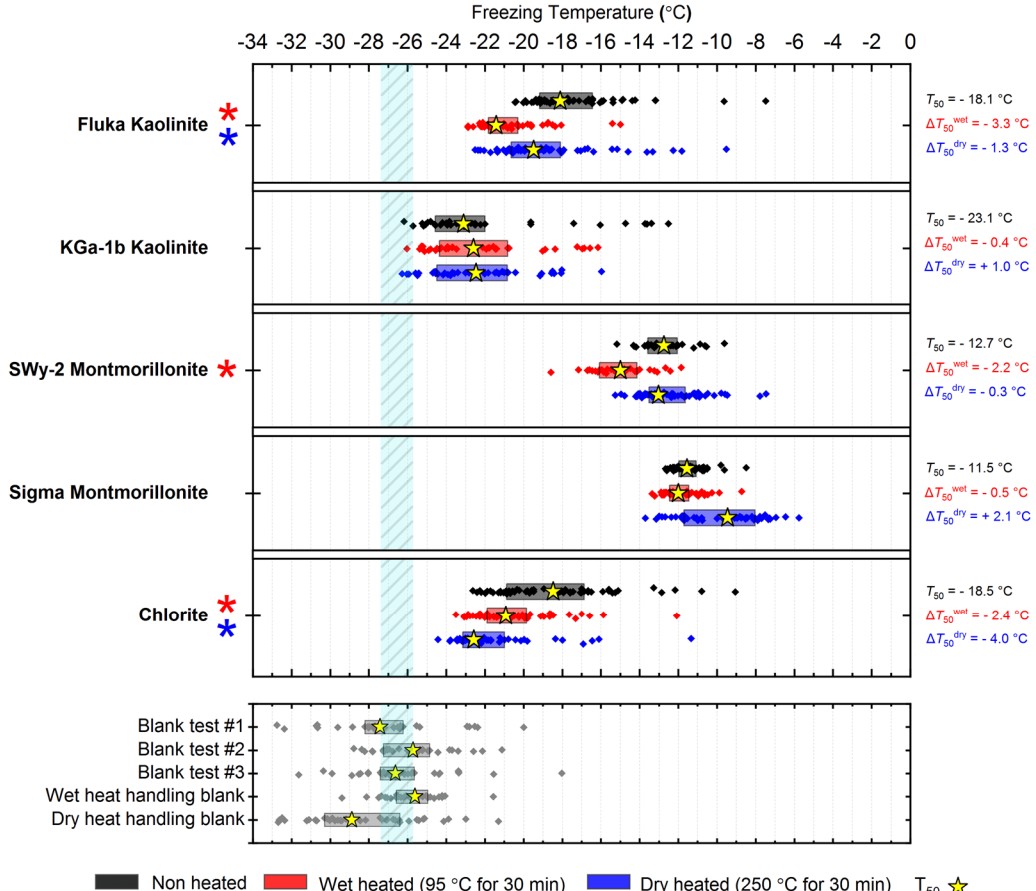

**Figure 5.** Boxplot showing freezing temperatures before (black) and after heat treatments (red for wet heat, blue for dry heat) for clay-based mineral samples.

The results for the montmorillonite samples were harder to interpret because both possessed quite low purities and showed responses to heat treatments that are not easily explained by their feldspar and quartz components. A notable result was that the INA of the Sigma montmorillonite sample increased after dry heat treatment. An increase in INA in the deposition mode after dry heating has previously been observed in a smectite-rich Saharan dust sample that had been dry heated at 300 °C for 10 h during a study by Boose et al. (2019). The authors discussed potential reasons for this, including the volatilisation and removal of an organic INA inhibiting coating, as well as purely inorganic processes such as the growth of new anhydrite crystals from gypsum in the sample or alterations to the lattice spacing of smectite clay. Our sample, which became more active upon heating, did not contain any gypsum impurities, hence conversion to the highly active anhydrite (Maters et al., 2020; Grawe et al., 2018) was not a possibility. Smectites are characterised by their ability to swell or shrink by taking up or losing loosely bound water molecules in the crystal lattice. However, the same effect of dry heating was not observed for SW-y2 Montmorillonite, which demonstrated no deactivation of INA with dry heating and only a minor deactivation upon wet heating. Hence, the changes in the INA of Sigma Montmorillonite during dry heating could not be related to its swelling properties.

The results for chlorite, with its high purity (99.6 %), indicated heat-lability in both the wet and dry heat modes. However, chlorite likely has only limited atmospheric importance as an INP due to both its relatively low INA

and typically low (around 5 %) proportional make-up of airborne mineral dusts (Murray et al., 2012; Kandler et al., 2009; Glaccum and Prospero, 1980).

### 3.1.5 Mineral dust analogues and calcite

The boxplots with droplet freezing temperatures for NX Illite and ATD are show in Fig. 6a. NX Illite was
unresponsive to wet heating, as was previously demonstrated by O'Sullivan et al. (2015), but deactivated after dry
heating with a $\Delta T_{50}^{dry}$ of −2.0 °C. ATD was clearly deactivated by wet heating ($\Delta T_{50}^{wet}$ of −4.5 °C, with activity
almost eliminated above −10 °C) and slightly affected by dry heating ($\Delta T_{50}^{dry}$ of −1.4 °C). In the case of both
these samples the responses were independent of suspension concentration (Appendix B). We further investigated
the wet heat sensitivity of ATD by performing extended heat treatments of up to 22 h and also room-temperature
ageing for 24 h, and these are plotted as $n_s(T)$ plots in Fig. 6b. This shows a similar behaviour to that observed for
Fluka Quartz (Fig. 4b), where room temperature deactivation was observed but deactivation was greatly increased
by heating. However, in the case of ATD, the sites active at above −10 °C retained activity when immersed in
room temperature water, while the activity of sites active at lower temperatures was reduced (Fig. 6b). A previous
instance of the wet heating of ATD in the literature also showed at a wet heat lability (Yadav et al., 2019), while
dry heating using a range of different methodologies also showed slight reductions in ATD's INA (Sullivan et al.,
2010; Perkins et al., 2020; Zolles et al., 2015), thus corroborating the results shown here. Perkins et al. (2020)
found that ATD lost some INA after dry heating to 500 °C, yet was deactivated to roughly the same degree by
'ageing' in water at room temperature for 2 days. Both dry heating to 600 °C and aqueous oxidation treatment by
boiling in 30 % $H_2O_2$ led to more significant deactivations. They attributed the dry heat deactivation to oxidation
of an organic coating stable in air up to 500 ˚C but removed readily in the aqueous mode. They did not, however,
boil the ATD in water alone which would have determined if the $H_2O_2$ deactivation was merely a result of being
heated in water.

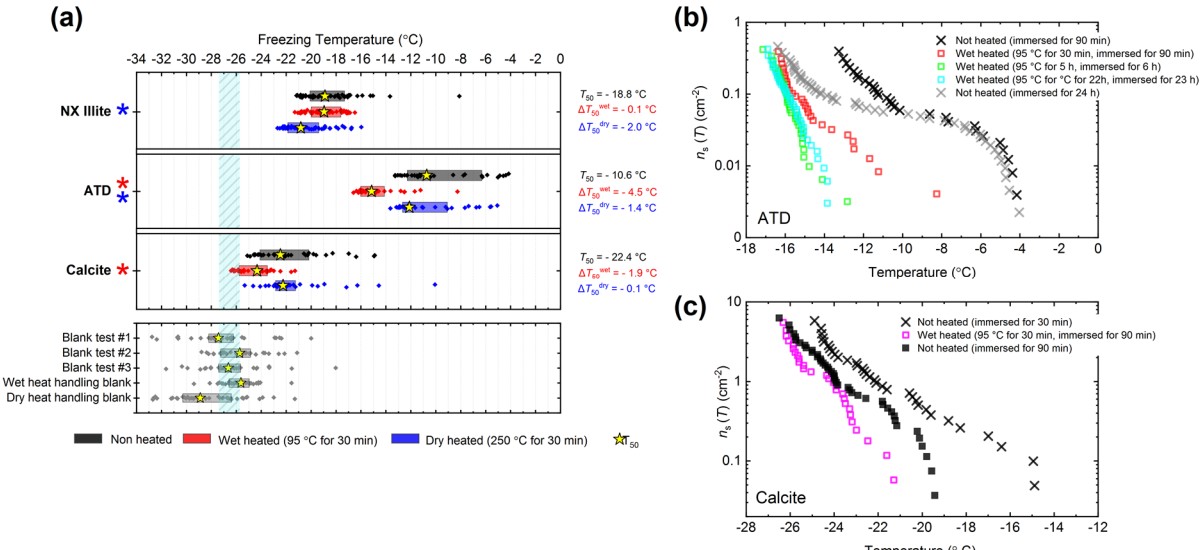

**Figure 6a.** Boxplot showing freezing temperatures before (black) and after heat treatments (red for wet heat, blue for dry heat)
for mineral dust analogues and calcite. **(b)** $n_s(T)$ spectrum for ATD after extended wet heat treatments and room temperature
ageing, illustrating their relative rates of INA deactivation. **(c)** $n_s(T)$ spectrum for Calcite after 30 min of wet heating and being
immersed in water for equal amount of time.

As described above, K-feldspar is mostly only sensitive to dry heating while quartz is only sensitive to wet heating, which implies that the observed changes in INA for NX Illite may be controlled by the K-feldspar component while the INA of ATD may be controlled by milled quartz particles. Alternative explanations to the deactivations include biological contamination. However, similar to the results obtained for the silica samples, the greater deactivation seen in ATD from wet heating compared with that from dry heating suggests that the heat-labile component is not biological.

The calcite sample displayed a reduction in INA after wet-heating ($\Delta T_{50}^{wet}$ of $-1.9$ °C) but not after dry heating. This case was distinct from that of other minerals that were sensitive to wet heating (e.g., Fluka Quartz) where a similar degree of deactivation occurred in a control experiment when suspended in water at room temperature for the same duration as the heated sample (Fig. 6c). If the dissolution of active sites on calcite resulted in INA deactivation in water, then the fact that heating did not significantly speed up INA deactivation can be explained by calcite exhibiting retrograde solubility in water. Dissolution of calcium carbonate in water occurs when water equilibrates with atmospheric $CO_2$ and forms weak carbonic acid. Hence, the solubility, and reduction in activity, is limited by the amount of $CO_2$ dissolved in water, hence the lack of additional effects upon heating.

**3.2 Biological INP surrogates**

Four biological INP analogue samples were subjected to the same wet and dry heat treatments (95 °C for 30 min and 250 °C for 4 h respectively). The results are summarised in Fig. 7 as boxplots of freezing temperatures, Fig B1e-h as $n_s(T)$ and $n_m(T)$ plots for samples over extended concentration ranges and in Fig. S2 as $f_{ice}(T)$ curves. We also performed wet and dry tests with varying durations and temperatures on Snomax® and birch pollen to compare the effects of heating in different media at equivalent conditions, these results are presented in Fig B2.

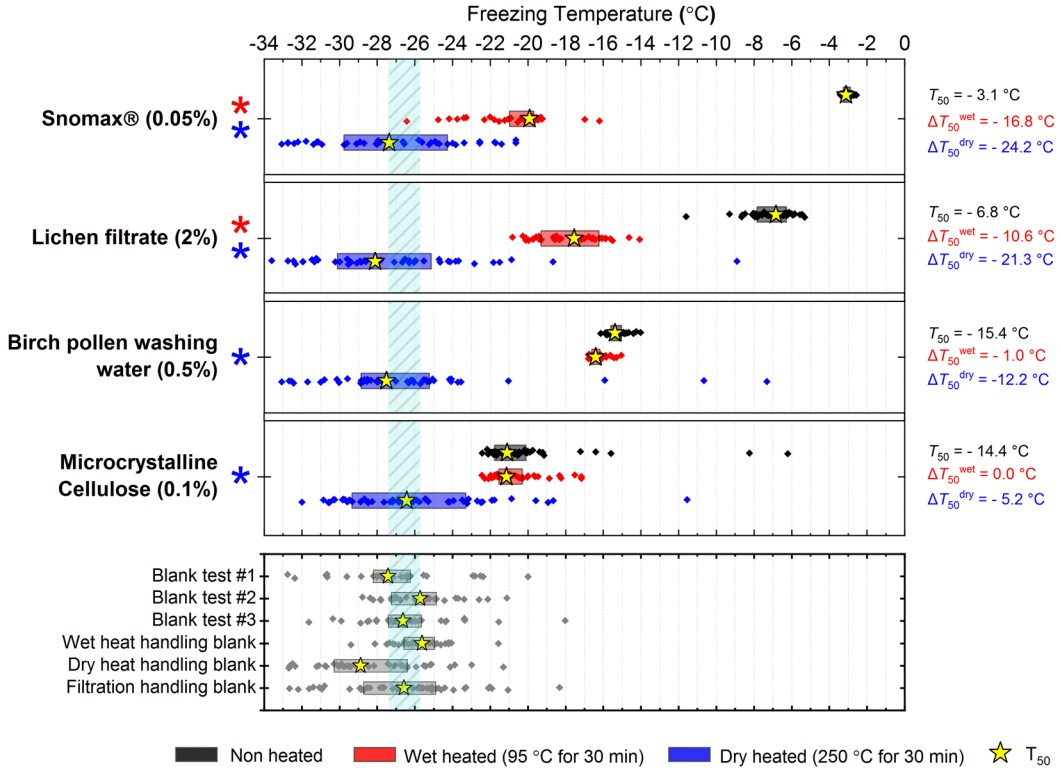

**Figure 7.** Boxplot showing freezing temperatures before (black) and after heat treatments (red for wet heat, blue for dry heat) for biological INP samples along with handling blank data for filtration procedure described in Section 2.2.

After the standard heat tests Snomax® (0.05 % w/v) was significantly deactivated by wet heating ($\Delta T_{50}^{wet}$ of -16.8˚C) and was also deactivated to background levels by the dry heat test with the material appearing carbonised (turning into a black substance) after the treatment. Clearly, both the wet and dry heats test denatured or destroyed ice-nucleating proteins in Snomax® although some residues with INA activity around -20 C° were left behind by

wet heat test. The activity was reduced to near background levels when the wet heating time was increased to 4 h (Fig B2b). Dry heating Snomax® at 250 ˚C for only 30 min instead of 4 h had the same results, with carbonisation and complete deactivation. Reducing the dry heat temperature to 95 ˚C only resulted in a very small deactivation (~1.2 ˚C, i.e. of borderline significance) after either 30 min or 4 h, with the Snomax® pellets appearing unchanged by the treatment. Lichen (2% w/v filtrate) showed similar behaviour to that of Snomax®, that of being carbonised

and deactivated to background levels by the dry heat test while the wet heat test achieved a significant deactivation but left residual activity between -15 ˚C and -20 ˚C, at slightly warmer temperatures than with Snomax®.

Birch pollen washing water (0.5 % w/v) was not significantly deactivated by wet heating ($\Delta T_{50}^{wet}$ of –1.0 C after 30 min and –1.6 ˚C after 4 h) but dry heating at 250 ˚C, resulted in deactivation of INA to background when

heated for 4 h and slightly above after 30 min. These wet heat deactivations were consistent when repeated with over 20- and 200-fold dilutions (Fig B2h). Dry heating at 95 ˚C for up to 4h, however did not result in any change to INA or any change in appearance of the raw pollen powder. Finally, microcrystalline cellulose (0.1% w/v) was

unchanged by wet heating (95 °C for 4h) but was completely deactivated by dry heating at 250 °C for 4 h and, like the other biogenic samples, was carbonised after this treatment.

Overall, the results showed that the bacterial and fungal derived samples (Snomax® and lichen) clearly suffered substantial deactivation by the standard wet and complete deactivation to standard dry heat treatments, while BPWW and MCC showed no or very little sensitivity wet heating but did to dry heating at 250 °C. For all four biogenic samples the magnitude of active site loss in terms of $n_m(T)$ and $n_s(T)$ was largely consistent when more

dilute samples where tested (Fig B1e-h) suggesting these tests are representative to a wide range of concentrations. The stability of the INA in pollen and polysaccharide-based cellulose heated in water is consistent with reports of the relative resistance of these particular ice-nucleating materials to similar treatment (Conen et al., 2011; Conen et al., 2015; Bogler and Borduas-Dedekind, 2020). We also corroborate results from Pummer et al., (2012) where Snomax® was able to be heated dry to 112 °C with only slight loss of activity. The relative stability of

proteinaceous INP heated to 95 °C in a dried state compared to in water is not surprising as it has been long known that the protein denaturation temperature is related to its water content (Barker, 1933). For example, using differential scanning calorimetry dehydrated proteins such as lysozyme (Phan-Xuan et al., 2021), soybean protein (Kitabatake et al., 1990) and collagen (Bigi et al., 1987) have been shown to be able to withstand dry heat of well over 100 °C without denaturing.

**4. Summary and implications for using INP heat treatments**

We performed both wet and dry heat tests on a range of mineral and biological ice-nucleating materials and directly compared their characteristic INA responses to both modes of heat treatment. Our findings, summarised in Table 4, show that the general assumption that the INA of minerals is insensitive to heat is too simplistic and we identified sensitivities characteristic to important mineral classes. For example, quartz and plagioclase feldspar

INPs were found to be sensitive to wet heating in a comparable way to proteinaceous INPs (bacteria and lichen), but were insensitive to dry heating at considerably higher temperature (250 °C). In contrast, K-feldspars are generally insensitive to the 30 min wet heat test (with the exception of Amazonite, which happens to be a relatively rare type of microcline), but slightly sensitive to the dry heating at 250 °C. Fluka Quartz and ATD, that were wet heat sensitive but not deactivated by dry heat, showed small INA deactivations after being dispersed in water but

kept at room temperature for less than 1 hr.. This suggests that the deactivation process for these samples is an aqueous process accelerated by increased temperature. Calcite showed similar behaviour in water except the room temperature and wet heat deactivations were similar. Of the other mineral samples we tested their interpretations were more difficult due to their INA possibly being controlled by impurities (Fluka Kaolinite with its quartz content, for example) but, overall, we found that mineral samples were more likely to be deactivated by wet

heating than by dry heating (with the important exception of K-feldspars).

The biogenic INP samples showed the clear heat sensitivity of bacterial and fungal derived INP - and heat resistance of pollen and cellulose INPs in wet mode, while dry heating at 250 °C served to eliminate all INA for both classes. Dry heating at 95 °C for 4 h did not deactivate BPWW and only slightly deactivated Snomax® in

contrast to wet heating at the same temperature and duration, which severely deactivated the latter and unaffected

the former. The magnitude of deactivations in terms of $n_s(T)$ did not depend on the concentration of INP during heating (Appendix B) nor the material of the vessel used (Appendix A). In the case of both mineral and biogenic samples longer heating duration led to greater degree of INA deactivation when wet heating but not when dry heating.

**Table 4.** Summary of the characteristic responses of classes of INPs to wet and dry heat treatments.

| INP type | Characteristic sensitivity of INA to wet heating (100 °C for 30 min) | Characteristic sensitivity of INA to dry heating (100 °C for 4 h) | Characteristic sensitivity of INA to dry heating (250 °C for 4 h) |
|---|---|---|---|
| K-feldspar | Stable[1] | Stable | Slightly heat sensitive[2] |
| Plagioclase feldspar | Slightly heat sensitive | (Stable) | Stable |
| Quartz | Heat sensitive[3] | (Stable) | Stable |
| Clays | Stable[4] | (Stable) | Stable[5] |
| Carbonates | Sensitive to water at room temperature[6] | (Stable) | Stable |
| Biological (bacteria and fungal) | Heat sensitive | Slightly heat sensitive | Heat sensitive |
| Biological (pollen and cellulose) | Stable | Stable | Heat sensitive |

Notes: (Stable) denotes assumed stability as heating to higher temperatures resulted in no deactivation ; 1. Hyperactive varieties may have slight sensitivity; 2. Hyperactive varieties are very sensitive; 3. Slight sensitivity in room temperature water; 4. Apart from chlorite; 5. Montmorillonite may increase in INA when dry heated; 6. INA deactivation does not increase when heated.

An implication of this work is that reduced INA of INP samples subject to a heat test may be incorrectly attributed to biological INPs when heated, particularly in wet mode. But crucially, since the INA of K-feldspar is not reduced by short-term wet heating, the standard wet heat test (30 min immersed in boiling water) remains a valid method for distinguishing bacterial and fungal proteinaceous INPs from mineral dusts, so long as the INA of the mineral dust component is controlled by K-feldspar. Nevertheless, the INA heat lability of some commonly occurring minerals raises the possibility that a false positive detection of biological INPs could be made following a wet heat test, i.e., a loss of INA of quartz or plagioclase feldspar may be misconstrued as a loss of biological INA. This could occur during a scenario in which a wet heat test is performed on a sample whose mineral component INA is dominated by its silica or plagioclase feldspar content rather than K-feldspar and in which bacterial and fungal proteinaceous biological INPs are absent. The importance of quartz and plagioclase feldspars as ice-nucleating components of mineral aerosols are second only to K-feldspar, hence the possibility of this scenario occurring should not be dismissed. However, feldspars and quartzes tend to be found together in desert dust assemblages, thus K-feldspar will likely control the INA of desert dust on most occasions.

Performing heat tests on minerals in parallel with biological samples allowed the magnitude of mineral wet-heat sensitivity to be put into context. For example, Yadav et al. (2019), performed wet heat tests on rainwater and dust samples collected from Northern India, with a heat test of ATD performed as a control. The results showed a resultant deactivation of INA that was consistent with our results. The authors attributed this to the presence of organic matter in their ATD sample. However, the magnitude of deactivation (~1 °C) observed in the ATD control was far smaller than in their rainwater samples (up to 10 °C), which was interpreted as evidence of 'biological influence'. In other words, the 'signal' produced by the mineral INP heat deactivation should be weak compared to that of bacterial and fungal proteinaceous INP deactivation, hence the loss of INA in ATD may not have been influenced by the presence of biological components after all. Generally, marginal heat deactivations of a few

degrees should be interpreted with caution and generally should not be attributed solely to the presence of bacterial and fungal proteinaceous ice-nucleating materials. This especially applies if heat deactivations have been used to calculate the ambient concentration of biological INPs in addition to identifying their presence.

We also consider the issue of whether the heat-sensitive active sites we found in our mineral samples are an artefact of the milling process and therefore not representative of particles present in the environment. The INA of quartz (Kumar et al., 2019a; Zolles et al., 2015; Harrison et al., 2019), hematite (Hiranuma et al., 2014) and also natural desert dusts (Boose et al., 2016) are increased by milling. This might imply that heat-labile mineral INPs do not occur naturally. Conversely, it has been argued that quartz particles in desert dusts are naturally 'milled' by collisions during the process of saltation prior to being lofted into the air (Harrison et al., 2019). If this is correct, then it would mean that only quartz INPs originating from desert dust, with their active surfaces exposed following saltation, would be wet heat-labile, whereas quartz particles that have been in contact with water, for example in soil or sediments, would have already been 'aged' and so may be less susceptible to further wet heat treatment.

Here, we provide some further caveats and considerations for the use of heat tests to identify biological, specifically proteinaceous, INPs in environmental and atmospheric samples:

*a) Dry heating INP samples as an alternative to wet heating*

Wet heating is the dominant mode of heat test used by others in past studies to detect biological (or more correctly, bacterial and fungal proteinaceous) INP (Table 1), yet wet heating deactivated some mineral INPs (e.g., quartz and plagioclase feldspars) much more strongly than dry heating did. Therefore, dry heating of aerosol filters or any INP sample available in an initially dry form could be considered as an alternative or parallel heating method that is more selective than wet heat treatment. Dry heating at 250 °C is expected to carbonise and deactivate all biological INP present, however the dry heat sensitivity of K-feldspar that we observed at this temperature could negate this approach. Our data shows that dry heating at a lower temperature of 95 °C preserved the activity of K-feldspar, however, it did not deactivate pollen nor cellulose and denatured bacterial and fungal proteinaceous INP (Snomax®) far less than when heated at the same temperature in water. This means that for detection of bacterial and fungal proteinaceous INP dry heating at 95 °C holds no advantage over wet heating at 95 °C. Bounded by the conditions tested here, further experimentation would help to determine if there is an optimal dry heating protocol that both preserves the INA of K-feldspar and deactivates all biological INP (or even selectively deactivates different types of biological INP). Ideally, this should be conducted using a combination of model materials like we used in this study, but also natural materials such as fertile soils, desert dusts, surface waters and precipitation samples. Some consideration also needs to be given to how a dry heat test would be conducted on aerosol sampled from the atmosphere. It may be possible to conduct a heat test on filters loaded with aerosol particles where filters could perhaps be split in two, one half to be heated and one half for the standard INP analysis. For this to be feasible, the effects and suitability of alternative dry heat protocols on aerosol filters should be investigated. Another potential approach might be to use an inlet system with a heated component that would heat aerosol to some specified temperature.

However, the timescales that aerosol would be exposed to elevated temperature would be relatively short (seconds) and tests would be needed to find the appropriate conditions.

*b) Optimise wet heat tests to avoid mineral deactivations*

As K-feldspar was seen to deactivate after prolonged (>30 min) wet heating, ensuring wet-heat tests are as short in duration as possible (i.e. no longer than the time it takes for the sample to reach, for example, 95 °C) and are then cooled and tested for INA as soon as possible would theoretically minimise wet-heat mineral deactivations. Also an overlooked consequence of the wet heat treatment is that the INP sample may be immersed in water for longer than non-heated counterparts. This could result in an apparent deactivation of non-biological INPs due to the room temperature 'ageing' effects of mineral INPs in water (as demonstrated in this study and in the literature (Harrison et al., 2019; Kumar et al., 2019a)), which we hypothesise are sometimes sped up by heating. Two adaptations to the method that could mitigate this include: (i) conducting tests on samples for room temperature 'ageing', similar to those for Fluka Quartz and ATD performed here; (ii) ensuring that all heated and non-heated samples are in immersed water for an equal duration before running tests for INA.

*c) Control heat tests on mineral and biological INP references*

Relatively few of the previous studies listed in Table 1, where heat treatments were performed to identify the presence of biological INPs in the environment, also included a control to test whether their protocol deactivated the INA of a reference material of known INA. Considering this study, future implementations of the heat test could benefit from testing a set of reference materials (e.g., microcline K-feldspar, albite plagioclase feldspar, quartz, Snomax® and pollen), to 'calibrate' a specific heat test protocol. Alternatively, the specific protocol described in this study could be used in future work.

**5. Conclusions**

In this study, we have tested and characterised the changes in ice-nucleating ability of the principal mineral components of desert dust in response to heat treatments in both wet and dry modes and in parallel with biological INP analogues (bacterial, fungal, pollen and cellulose). The main purpose of this was to assess the efficacy of heat treatments for the 'detection' of biological INPs in environmental sample media such as ambient aerosol, surface waters, soils and desert dusts. Understanding how the sources and distribution of biological INPs and mineral dust INPs differ in the environment may be crucial for understanding their current and future impact on the climatic impacts of clouds. It has been previously assumed that mineral INPs are inert to moderate heat treatments that are sufficient to denature proteins. However, we found that while the INA of (most) K-feldspars was unchanged on wet heating for 30 min, as expected, quartz and plagioclase-rich feldspars were heat-labile. The INA of quartz and plagioclase-rich feldspar samples was unchanged when exposed to dry heat (250 °C for 4 h). Given that all biogenic INP samples were strongly deactivated by the dry heat test, it is clear that the loss of activity in quartz and plagioclase feldspars was related to the minerals themselves, rather than some biological contamination.

We suggest that the loss of INA on wet heating of quartz and plagioclase feldspars is related to aqueous dissolution of features acting as active sites on the mineral surface. This is supported by the observation that the relative dissolution rates of the different mineral types correlate with their relative heat sensitivities. Moreover, several

studies have previously reported aqueous room temperature 'ageing' of mineral INP samples and our results are consistent with the same process being accelerated by heating. As quartz and plagioclase feldspars are ubiquitous components of mineral dusts, this raises the possibility of false positives being produced by minerals in wet heat tests, which are more commonly used compared to dry heat tests. However, if the mineral-based INA of an environmental sample being tested for INP is controlled by K-feldspar then wet heat tests are valid.

Dry heating produced stronger deactivations compared to wet heating in the biological INP analogues, while overall being less likely to deactivate minerals. This could mean that dry heating has less potential to produce false-positive detection of biological INPs, so could be a more appropriate method for INP heat tests since wet heating is the method usually employed in these investigations. However, this may be precluded by the finding that most of our K-feldspar samples exhibited dry heat deactivations. Due to its practical simplicity and potential for high throughput of samples, heat treatments will likely continue to be the primary method used in future studies where biological INPs need to be differentiated from other types present in a collected sample. Interpretation of results may by aided by identification of the mineral phases present in a sample using techniques such as XRD or SEM. Overall, we have highlighted potential limitations where INP heat tests are applied and the need for deeper interpretation of results and have outlined possible improvements to INP heat treatment methods. Further studies should focus on finding the optimum physical conditions that would result in the most selective deactivations of biological INPs.

**Appendix A:** The effect of vessel type used for wet heating of silica INP suspensions.

In our wet heating experiments, the mineral INP suspensions were heated while inside 20 mL borosilicate glass vials containing 10 mL of suspension. Kumar et al. (2019a) observed that ageing of quartz INP suspensions over several days occurred at room temperature in glass vials but not in polypropylene centrifuge tubes. For this they proposed an alternative explanation to the active sites on the quartz INP being irreversibly degraded by ageing in water, in which silicic acid leaches out from the glass vial walls and re-precipitates onto the active sites of the mineral, effectively blocking them. When polypropylene is the suspension container, however, the Si concentration remains too low for this to occur, so the INA does not reduce. Therefore, to rule out that quartz wet heat deactivations are only an artefact of heating in glass containers, we repeated our wet heat test for Fluka Quartz in an alternative glass vessel type (20 mL non-borosilicate vial with 10 mL of INP suspension) and plastic vessels (50 mL propylene centrifuge tube and 1.5 mL polypropylene microcentrifuge tube (Sarstedt Micro Tube 72.690) containing 10 mL and 1 mL of INP suspension, respectively) and compared the deactivations with those seen for our standard wet heat treatment in borosilicate glass. The results in $f_{ice}(T)$ from are shown in Fig A1.

Similar or larger wet heat deactivations occurred for the 1.5 mL microcentrifuge tube ($\Delta T_{50}^{wet}$ of −6.0 °C) and non-borosilicate glass vial ($\Delta T_{50}^{wet}$ of −4.8 °C) samples compared to that of borosilicate glass vial ($\Delta T_{50}^{wet}$ of −4.1 °C). With the 50 mL polypropylene tube sample, however, a smaller deactivation occurred: $\Delta T_{50}^{wet}$ of −1.8 °C. This may simply be because the suspension in the polypropylene tube did not reach as high a temperature as that inside the glass vials while immersed in the water bath, as demonstrated in Fig. A2 based on temperature measurements in both vessel types during the heat test procedure. This may have been due to the thicker wall and lower thermal conductivity of the polypropylene tube compared to the glass vials and microcentrifuge tube.

Nevertheless, deactivation of quartz INP was still achieved when suspension were heated within plastic tubes, suggesting that aqueous silica leached from a glass container does not play a role in the deactivation of INA.

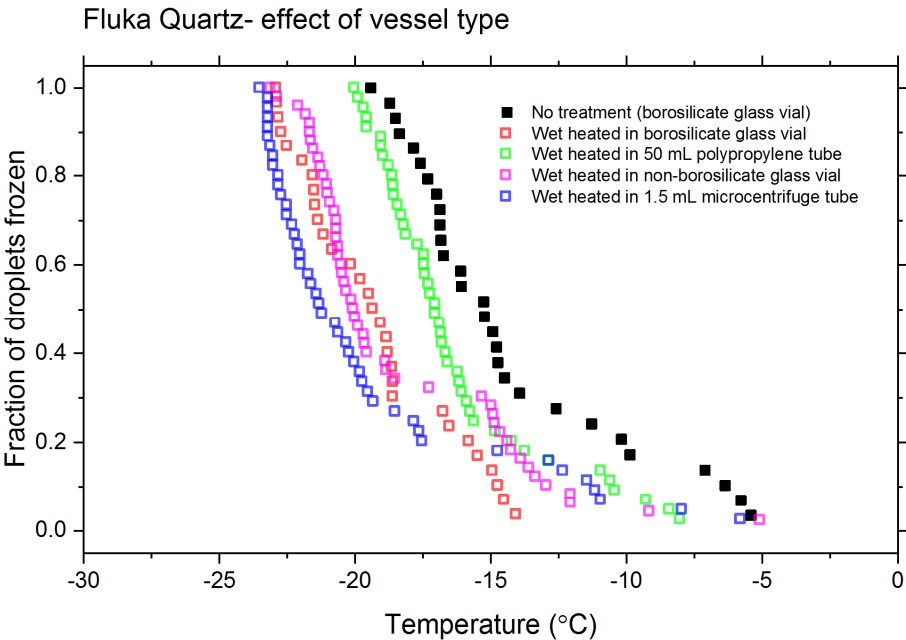

**Figure A1**: Plot showing the fraction of droplets frozen ($f_{ice}(T)$) for wet heated Fluka Quartz suspensions (all 1 % w/v) repeated using a range of different container types.

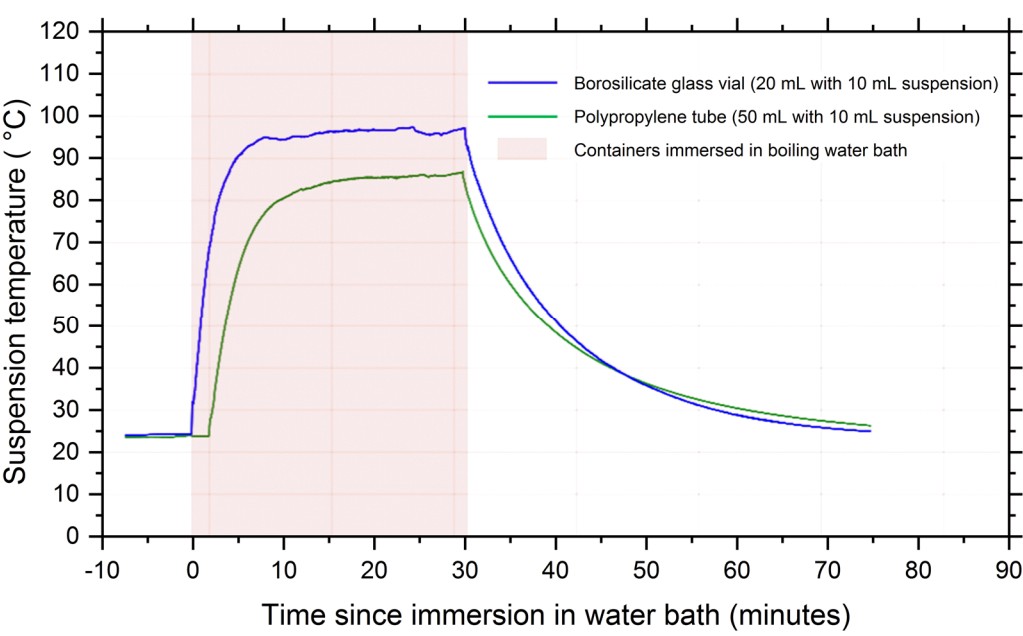

**Figure A2:** Thermocouple measurements of suspension temperature inside both glass and plastic vessels during the wet heat treatment procedure.

**Appendix B:** Dependence of INA heat deactivations on temperature, duration and suspension concentration

Several mineral samples' INA were significantly deactivated by heating in this study and we hypothesise that their ice-active sites are degraded by elevated temperature but also dependent on whether the mineral samples were heated while immersed in water or dry in air. Here we explore the dependence of additional variables on heat treatments a using some of the mineral INP samples and the biogenic INP samples included in this study.

We used relatively concentrated suspensions of minerals and biogenic INP for the experiments shown in section 3 in order to ensure their droplet freezing temperatures were well above the instrumental background. However, there are potential mechanisms for the concentration of the suspension itself to affect the INA independently from the species dissolved from the mineral powders may potentially interact with the

nucleation to reduce the INA (Koop and Zobrist, 2009; Kumar et al., 2018a; Whale et al., 2018). Agglomeration of particles causing loss of INP surface area has been proposed to cause lower than expected INA with increasing INP concentrations (Emersic et al., 2015; Hiranuma et al., 2019). Also, more concentrated suspensions are more likely to contain rarer, warmer temperature IN sites which may be of different nature, and thus, different heat sensitivity to lower temperature IN sites. We therefore repeated both wet and dry heat tests for BCS-376

Microcline, Fluka Quartz, NX-Illite and ATD at both higher and lower concentrations than the standard 1% w/v reported in Section 3. This allowed us to ascertain if the observed heat deactivations are an artefact of the relatively concentrated suspensions that we used and are still pertinent to the lower particle concentrations involved with, for example, aerosol filter wash-offs. We also performed this with the biogenic samples for wet-heat tests only, since dry heat tests deactivated all these samples to background levels.

The resultant droplet freezing data is plotted in $n_s(T)$ and $n_m(T)$ form and shown in Figs. B1a-i. Plotting $n_s(T)$ or $n_m(T)$ data for the same INP sample repeated at multiple concentrations should result in a coherent 'curve' with data for lower concentrations reaching into higher values of $n_s(T)$ (or $n_m(T)$) and vice-versa. This is indeed the case for all samples we tested whether they were sensitive to wet heat (e.g. Fluka Quartz, Lichen) or dry heat (e.g

BCS-376 microcline, NX Illite) (Fig B1). If, however, there was a concentration dependence on INP deactivations then the $n_s(T)$ data for wet heated suspensions would be 'staggered' rather than coherent and would not be parallel with the unheated sample's $n_s(T)$ curve. In Fig. B1 we see that for all samples, for both wet and dry heating, the $n_s(T)$ or $n_m(T)$ curves are largely coherent and parallel with those for the unheated data showing that the rare, high temperature active sites are equally as heat sensitive as the more abundant low temperature sites.

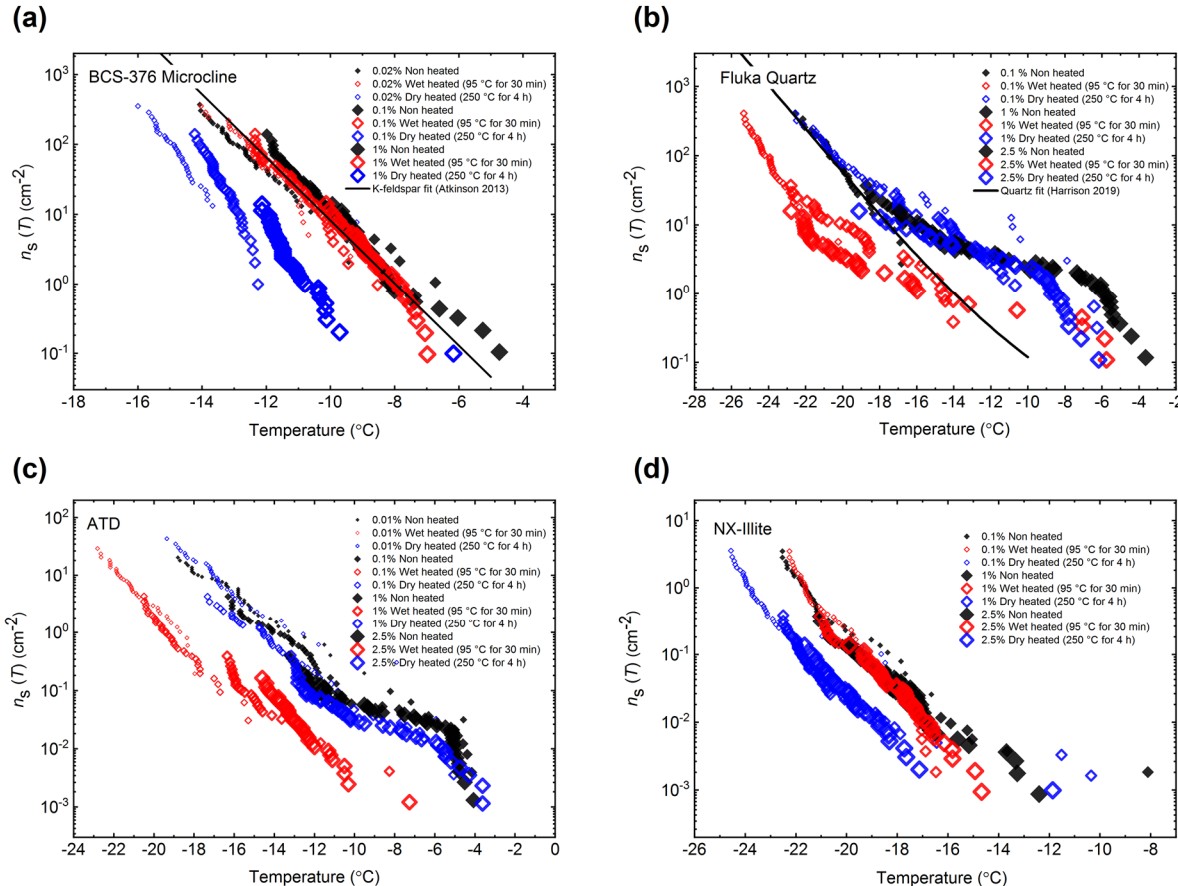

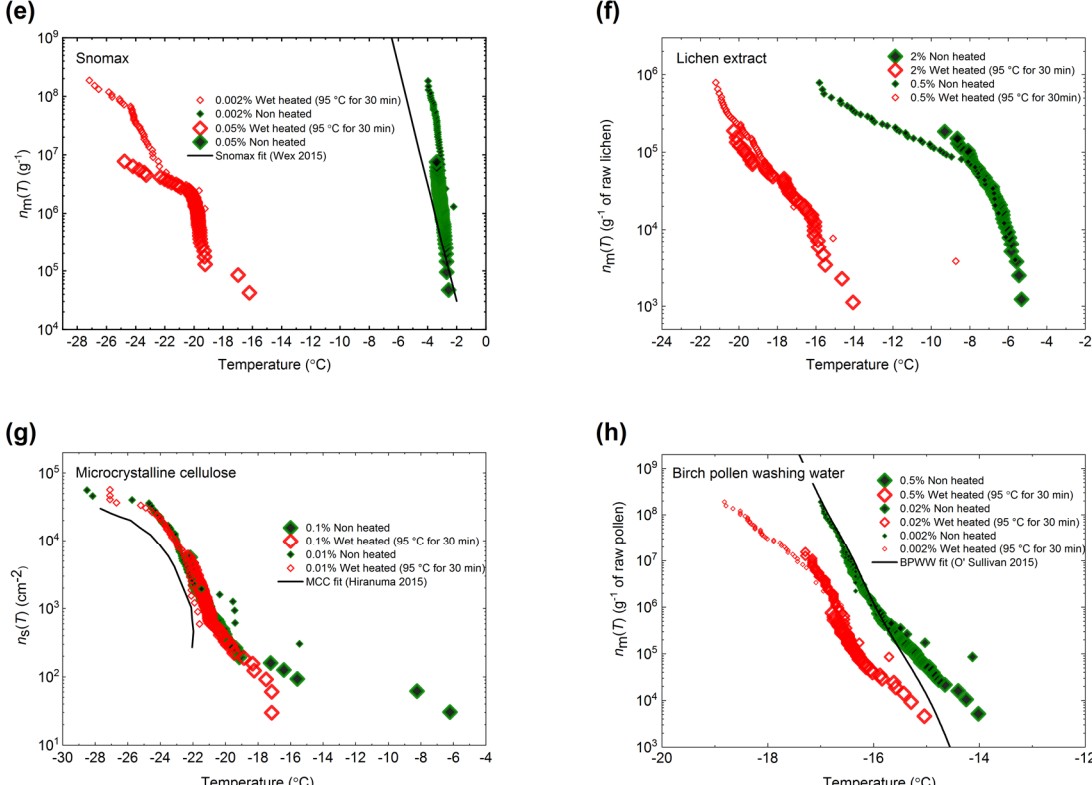

**Figure B1** Plots of $n_s(T)$ or $n_m(T)$ illustrating heat test responses over an extended range of suspension concentrations for a) BCS-376 Microcline, b) Fluka Quartz, c) ATD, d) NX-Illite, e), Snomax® f) Lichen extract g) MCC and h) Birch pollen washing water.

In section 3 all samples were subject to 'standard' heat treatment conditions of 95 ˚C for 30 min for wet heating and 250 ˚C for 4 h for dry heating. We did this to empirically test these two distinct heat test procedures, but from a mechanistic perspective these experiments are not ideal since we are varying heating mode, temperature and duration. Hence, we performed a set of experiments with a subset of samples (BCS-376 Microcline, Snomax and Birch pollen washing water) as follows: Dry heat at 95 °C for 30 min, 95 °C for 4 h, 250 °C for 30 min and wet heat at 95 °C for 4 h. These results are shown in Fig B2. Fluka Quartz is also included but without extra dry heat tests as it did not deactivate after dry heating at 250 °C for 4 hr so we assumed it would not deactivate if dry-heated at a lower temperature and/or for a shorter duration. Overall this allowed us to conclude that heating duration is a more important variable when wet heating than in dry heating where it appears secondary in importance to temperature. Also, Snomax, and to a lesser extent BCS-376 microcline, showed differences in their response to being heated wet and heated dry at the same duration and temperature.

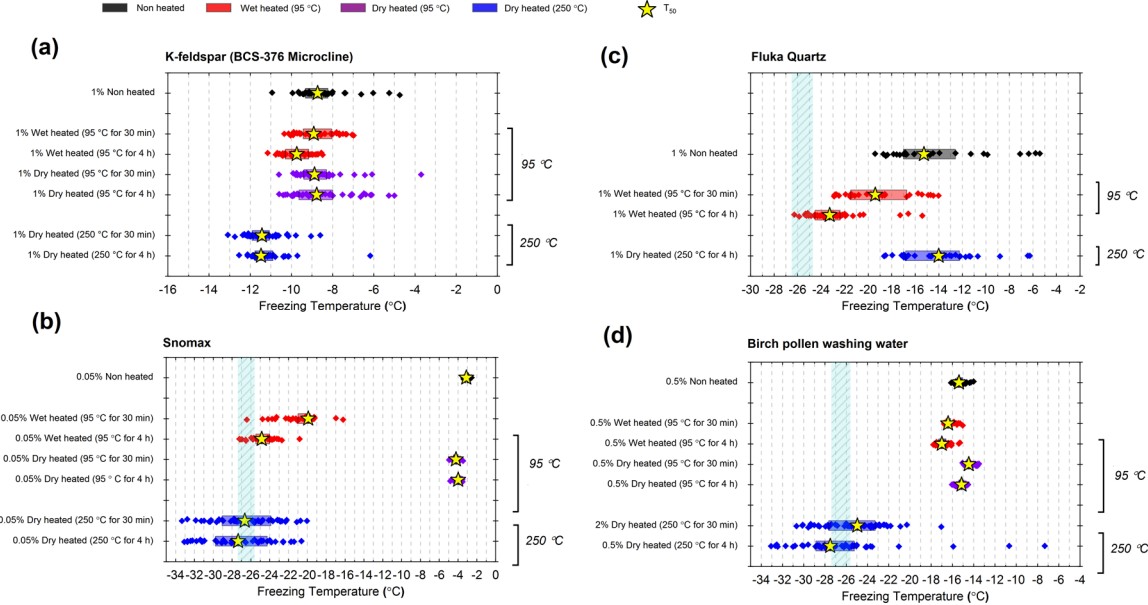

**Figure B2** Boxplots of freezing temperatures for droplet freezing assays of suspensions of a) BCS-376 Microcline (1 % w/v) , b) Fluka Quartz (1 % w/v), c) Snomax® (0.05 % w/v) and d) Birch pollen washing water (0.5 % w/v) before and after wet heating and dry heating at varying temperatures and durations.

*Supplement.* The supplement related to this article is available online at: https://doi.org/xxx/xxx/xx-supplement.

*Data availability.* The dataset for this paper, including raw droplet assay freezing data, is publicly available at the University of Leeds Data Repository - https://doi.org/10.5518/1002 (Daily et al., 2022).

*Author Contributions.* The study was conceptualised by MID and BJM. MID designed and performed the experiments with scientific input from BJM and TFW. MID prepared the manuscript with contributions from all co-authors.

*Financial support.* This research has been supported by the Natural Environment Research Council (grant no. NE/L002574/1), the European Research Council (ERC, MarineIce: grant no. 648661), along with Cytiva (formerly Asymptote Ltd), Cambridge, UK. TFW thanks the Leverhulme Trust and the University of Warwick for supporting an Early Career Fellowship (ECF2018-127)

*Competing interests.* The authors declare that they have no conflict of interest.

*Acknowledgements.* The authors are grateful to Alex Harrison and Jim Atkinson who originally sourced many of the mineral samples used in this study and to Mark Holden who provided the Amazonite and Pakistan Orthoclase samples. Ulrike Proske collected and preserved the lichen sample and performed initial measurements of its INA. Andrew Hobson and Andrew Connelly provided valuable laboratory support.

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
