# Peer review of "An evaluation of the heat test for the ice-nucleating ability of minerals and biological material"

_Atmospheric Measurement Techniques, 2021_

## Author Comment (AC1)

Response to Reviewer 1 Comments.

We would like to thank the reviewer for their constructive feedback and suggestions. We present our responses and the resultant changes to the original manuscript below. We have re-numbered the reviewers' comments (R1 or R2 C#1, C#2 etc.) and split some up (a, b, c etc.) in order to respond to individual points. Our responses are written below in blue 10pt text with **changes indicated by bold type.**

General comments:

The authors heated 20 samples of minerals (K-feldspar, plagioclase feldspar, quartz, clay, dust surrogate and carbonate) as well as 4 samples of biological material (Snowmax, lichen, birch pollen washing water and cellulose) using two different heat test procedures. The first was a wet heat test where the samples were submerged in a bath of boiling water and the second was a dry heat test at 250 °C in an oven for 4h. The authors then measured the samples before and after the two types of heat tests and display their results as frozen fractions, as box plots and as n_s plots. The authors then have rather long discussions of speculations (the word assume/assumption is found at least 10 times in the manuscript, at times justified and at times not, when the assumption could be resolved with further experiments). They speculate about what could have driven the differences before and after heat for the different heat tests and for the different types of samples. In general, the authors reference the literature adequately and thoroughly.

The research question is certainly worthwhile, and the authors' systematic approach is a good idea for evaluating the general applicability and the interpretation of a change in INA after a heat test. I commend the authors for approaching this problem systematically. However, this manuscript is currently too preliminary to be published. This study can be made significantly stronger to make an impact on the community and for the work to be built upon in the future. My key recommendations to improve the study before publication are below:

**R1C#1**. (most important recommendation) The authors are missing key experiments for further conclusions to be drawn. Specifically, the authors should run all 20 mineral and all 4 biological samples in a comparable dry heat test at 95 °C in an oven for 30 mins (or the same amount of time the sample was submerged in the water bath). This test is necessary, since the authors make many assumptions of what can be the cause of inconsistencies between their wet heat test and their dry heat test. Yet, comparisons have 3 variables being changed in both sets of heat tests: the method, the temperature and the time of heating. I would also encourage the authors to consider re-runing their dry heat test at 30 mins (or running all their heat tests at 4h), which would add additional columns in Table 3. These additional experiments would really strengthen the systematic approach that the authors are attempting to present in this work. Right now I am left wondering what is the effect of wet vs dry and what is the effect of temperature and what is the effect of heating time?

There are effectively four variables we have investigated here: mode of heating (wet vs dry), type of INP, duration and temperature and so have tried to strike balance between all four of these. The heating conditions we used for both wet and dry modes were chosen on the basis of being representative of those typically used in the literature (refer to Table 1 in the manuscript and paragraph directly below). Instead of varying the heating conditions we chose to test a large array of

mineral samples due to the diversity of atmospherically relevant minerals groups and the variability of INA seen between samples of the same mineral type. Time and resources limit us to a finite number of tests we can carry out and as such the full scope of experiments suggested within R1C#1 is unrealistic - around 50 additional droplet freezing assay runs from this comment alone. Furthermore, the key objective of this study was to test the validity of the key assumption made throughout the community that the INA of minerals is insensitive to heat and the experiments we presented are well-suited to doing achieving this goal.

However R1 makes a valid criticism that we contrasted the different heating modes (wet/dry) at different temperatures and durations. **We therefore carried out further heat tests on a selection of samples – BCS-376 Microcline, Snomax and Birch pollen washing water: Dry heat at 95 °C for 30 min, 95 °C for 4 h, 250 °C for 30 min and wet heat at 95 °C for 4 h. These new results are shown in Fig B2 in Appendix B.**

We made the reasonable assumption that samples that did not deactivate after dry heating at 250 °C for 4 h would also not deactivate if dry-heated at a lower temperature and/or for a shorter duration. Therefore, we did not perform extra dry heat tests on other mineral groups such as quartz, plagioclase and the clays. **An additional wet heat test on Fluka Quartz at 95 °C for 4 h is shown in fig 4b.** We also point out that 4 h/95 °C wet heat tests were in the original manuscript for ATD (fig 4b, now 6b) and BCS-376 Microcline (fig 2b).

Overall, this allowed us to conclude that heating duration is a slightly more important variable when wet heating than in dry heating where it appears secondary in importance to temperature. Also, Snomax, and to a lesser extent BCS-376 microcline, showed differences in responses to being heated wet and heated dry for the same duration and temperature. These extra tests, and the assumption that samples resistant to dry heat at 250 °C are also resistant at 100 °C, allowed us to **add a column to Table 3 in Section 4 showing the characteristic responses to dry heat tests at lower temperature of 100 °C and also allowed us to refine our conclusions and recommendations sections.**

**R1C#2a**. Furthermore, there are important background water tests missing in this manuscript. The authors should address these details thoroughly before publication. In general, daily blank tests discussed in lines 261-262 are not experiment controls, and do not represent adequately the experimental procedure each sample is submitted to. The authors use 0.1 μm pre-filtered, cell culture-grade deionised water, and could the authors show the following background water data:

A sample of background water that was heated in the same type of vial as with the wet heat test (line 208).
A sample of background water that was heated in the oven at 95 °C in an oven for 30 mins.
A sample of background water that was passed through the same experimental procedure following the sample after heating at 250 °C in an oven for 4h.

**A new plot ($f_{ice}$ vs T) with handling blanks and five background water tests is shown in Fig S3 in the Supplementary Info.**

**R1C#2b**. A sample of background water that passed through the nylon net filter and the cellulose acetate filters (referred to in line 192). I highly suspect that cellulose acetate filters leach material. These control frozen fractions must be shown.
Show the data discussed in lines 221-224.

We are grateful for R1 for suggesting this because it revealed suspected contamination of the stainless steel filter holder (Advantec 301000) we used with the nylon net filter from the birch pollen washing water (BPWW). We believe this is the case as i) The handling blank we did using the same procedure produces a steep signal at around –19 °C but did not before the original experiments. This is despite thorough washing of all parts (PTFE gaskets, metal screens) in isopropanol and deionised water. ii) Handling blanks of the nylon net and cellulose acetate filters (CA) used separately did not produce this steep signal.

For this reason, we adapted the filtration method in section 2.2 and repeated all experiments with birch pollen washing water and lichen using a protocol which did not involve the stainless steel filter holder. We did this by taking the raw suspensions from the vial with a sterile 1 mL syringe and then pushing them through a disposable 0.2 µm cellulose acetate filter (Sartorius Minisart). The raw suspensions were pushed through with no apparent resistance and resulting filtrates were crystal clear, suggesting the 10 µm nylon net filter was not needed. The handing blank for this process (Fig S3) shows the CA filter raised the background slightly but this was avoided by pre-flushing it with 10 mL of deionised water.

**As a result of these adaptations, we found that the INA of the BPWW and lichen extracts are indeed reduced to background levels by the dry heating (250 °C), in line with previous works (Pummer et al., 2012). This allowed us to update the figures and conclusions accordingly, as well as the summary in Table 3.**

**R1C#2c**. What is the role of the pre-sterilized dry heating at a different temperature? Shouldn't the procedure also involve the same 250 °C temperature as the experiment?

This was following a general protocol for sterilising glassware stored in ambient air, rather than one for specifically destroying contaminating INP. **A dry-heating handling blank at 250 °C is shown in Fig S3** and this shows that this treatment reduces the background somewhat compared to the blanks 1-5 which were pre-sterilised at 175 °C. However, the vast majority of experiments produced data well above this background anyway so pre-heating at 250 °C would have been only marginally beneficial.

**R1C#3**. The wet heat test will certainly have the water evaporate and therefore change the concentration of the material within the solution. How are the authors accounting for changed in concentration of the ice active material? relevant to the discussion in lines 145-147.

Refer to Section 2.3.1 (Wet heating) "The vessels were sealed tightly to prevent the evaporation of water from the vessel causing an increase in concentration of the suspensions". We tested this by weighing vessels before and after a wet heat run and the mass was unchanged within 0.1g. We also ensured the water bath was never high enough to reach near the lid and potentially leak into the vessel.

**R1C#4a**. The authors should make every effort to compliment the study with alternative measurements. I can appreciate that substantially more work would be required, but it would allow the manuscript to be much more concise rather than listing a list of speculations (for example, the K-fledpsar discussion spans pages 11-14 of speculations). For example, ideas presented in lines 365-377 could be address with elemental analysis and presence of N. Or with a protein test such as the Lowry method.

The text the referee refers to is not simply speculation.  We think the majority of this is a robust discussion of  the mechanistic reasons behind the observed heat test responses. For example, in the case of the hypothesis for the wet-heat deactivations of quartz and k-feldspar, the ideas are backed

up by our experiments and by existing literature. **We have removed the discussion on lines 379-387 of the original manuscript as it is the most speculative out of all the ideas presented, and have replaced this with a discussion about coatings blocking active sites on K-feldspars in light of findings by Pach et al (2021).**

Regarding suggestions for extra chemical analysis, we re-iterate that the core motivation was to empirically test the heat sensitivity of mineral samples. The further work that the referee suggests will form the basis for future studies.

**R1C#4b**. Ideas in lines 368-370 could be tested with a heat test at the same temperature as the wet test (see my point number 1).

See comments above (R1C#1)

**R1C#4c**. In addition, total mass of the material after both heat tests can be weighed and measured (assuming the wet heat test and be dried out).

**We did this for the dry heat tests (see Supplementary Table 1) and found that the change in mineral samples' mass did not correlate with dry-heat deactivation**. Weighing after wet-heating would be problematic as drying them would require, in-effect, dry-heating them.

**R1C#4d**. Have the authors attempted to dry their material and repeat the heat tests multiple times? Does a wet heat test followed by a dry heat test and vice versa have any effect?

We have not performed these tests and while they would be interesting supplementary experiments they do they do not fall within the primary goals of the study which is to evaluate the common heat tests methodology used..

**R1C#4e**. Evidence of chemical composition would also be particularly helpful, for example total organic carbon analysis, ion chromatography, elemental analysis, etc. I'll add here that my criticism is also a general one to our community where we tend to only show INA of material when these results cannot simply be compared without other measurements (such as surface area and/or composition).

The majority of mineral samples we used in our study have featured in previous studies about their ice-nucleating properties and as a result a wealth of compositional information is available in the literature. Using this we compiled values for specific surface area and compositional (purity) for most of our samples in Table 2, Section 2.1. We accept that some of these baseline properties may change after our heat treatments, but we reiterate this falls outside the scope of this study (which specifically INA responses to heat) but could form the basis of a future study focused on a smaller number of samples.

**R1C#4f.** Finally, point d) at lines 719-725, I would recommend that the authors either don't discuss it, or do the experiments.

**We accept this falls outside of the stated scope of the study and have removed this passage.**

**R1C#5**. The authors should further describe their detailed storage protocols. These protocols come up a few times as excuses for differences but should be detailed to teach the community exactly what was done. (Examples include Line 166, 195 and 201). I'll also add that the authors' justification on lines 234-235 is very good.

**We have added "The biogenic samples were stored cold (Snomax at -20 °C, raw birch pollen and dried lichen at – 4° C) or at room temperature in the case of MCC and they were made to suitable concentrations according to existing literature protocols"**

**And "All mineral samples were stored at room temperature in darkness and suspensions were prepared by mixing 0.1 g of sample with 10 mL water in 20 mL borosilicate glass vials (Samco type T006/01, Surrey, UK).**

**R1C#6a**. The authors use a high concentration of material (stated as 20 mg/mL on line 190).

We accept the concentrations we used were rather high in the case of Snomax, BPWW and MCC (see response to R2C## below**) so we have therefore re-done the experiments at maximum concentrations of 0.05%, 0.1% and 0.5%, respectively.** We found that Snomax did not show residual INA after a 30 min wet heat test and was more thoroughly deactivated. BPWW and MCC at lower concentrations did not appear to behave differently to the original experiments done at higher concentrations in terms of $n_m(T)$ values or their heat responses.

**R1C#6b**: Are the authors working with suspensions or homogeneous solutions (line 230)?

Although we generally have referred to all samples as suspensions throughout, **we have clarified this point in the case of BPWW and lichen in section 2.2: "The birch pollen and lichen samples could not be immediately dispersed in water as they required additional filtration steps to produce visibly clear homogeneous extracts rather than particulate suspensions."**

**R1C#6c**. This fact is important since after heating, material's solubility can substantially be affected. How have the authors addressed a possible change in solubility before and after their heat tests? What is the effect of concentration on the heat test? I think a series with one type of mineral sample with different concentrations for both heat tests would be an interesting series to present.

**We have addressed this by carrying out additional runs at varying concentrations in addition to their 'standard' concentrations, before and after both standard wet and dry heating for:**
**BCS-376 microcline (0.1% and 0.02%)**
**Fluka Quartz (2.5%, 0.1%)**
**ATD (2.5%, 0.1%, 0.01%)**
**NX Illite (2.5%, 0.1%)**
**Snomax (0.002%)**
**BPWW (0.02%, 0.002%)**
**Lichen (0.5%)**
**MCC (0.01%)**

**We then plotted the results in $n_s(T)$ form or $n_m(T)$ where appropriate in Fig B1a-h. In all cases, the $n_s(T)$ or $n_m(T)$ curves, both heated and non-heated are coherent indicating there is no concentration dependence. This is explained fully in Appendix B and referred to in the main text where appropriate. Note there is no data for dry heat tests for the biological samples as they were all already fully deactivated at the highest concentrations.**

**R1C#7**. Can the authors show what the role of the time during the heat test can have? This information would be particularly helpful to determine and optimal temperature and time of heating for subsequent experiments by the community (and will be better cited).

**This is addressed with new data presented in Figure B2 and discussed in the final paragraph of Appendix B, (see also R1C#1) where we performed heat treatments on a subsection of samples at varying temperatures and durations**. The general conclusion was that when we varied heating duration, longer wet heat tests resulted in further deactivations whereas dry heat tests showed similar results regardless of duration.

**R1C#8**. Have the authors attempted to combine one of their mineral samples with one of their biological samples? This test would better represent an ambient measurement and see if the effect is cumulative or not (see (Steinke et al., 2020)).

This would be very interesting but to gain meaningful insight it would require several combinations of mineral and biological samples with differing heat resistances, heated both wet and dry. We believe this would be better placed in a further study where it could be compared with natural samples such as fertile and desert soils.

We have added "Ideally, this should be conducted using a combination of model materials like we used in this study but also natural materials such as fertile soils, desert dusts, surface waters and precipitation samples. Also, the effects and suitability of alternative dry heat protocols on aerosol filters should be investigated" to acknowledge this point.

**R1C#9.** References: In general, Table S1 could be included in the main test as a reader-friendly reference guide for future work to compare and built upon. Good job to the authors for this compilation - although mentioned in the text, did the authors want to also include (Bogler and Borduas-Dedekind, 2020) in their table S1?

We have moved this table to the main text (Table 1) and updated it to include Hiranuma et al. (2021), Zinke et al. (2021) and also Tobo et al. (2014) which was omitted. The table is intended to highlight the heat treatments used for samples collected in the environment rather than heat resistance of particular ice-nucleating materials which is summarised in Section 1, paragraphy below Table 1. Bogler and Borduas-Dedekind (2020) was therefore not included as they did not perform their heat tests on environmental samples, rather on reagent grade ligin.

**R1C#10**. Title: Seems to me that the second part of the title is most relevant to the content of the work. I can encourage the authors to consider a title along the lines: Testing (or systematic evaluation) of the heat test for ice-nucleating ability of minerals and biological materials.

Changed to "An evaluation of the heat test for the ice-nucleating ability of minerals and biological material".

**R1C#11**. The introduction can be substantially shorted to focus only on the heat tests experiments listed in Table S1.

Lines 74 –90 of the original manuscript have been condensed to the following in paragraph 7 of Section 1 (Introduction):

"While techniques such as genomic sequencing (Garcia et al., 2012; Huffman et al., 2013; Hill et al., 2014; Christner et al., 2008) and microscopy (Huffman et al., 2013; Sanchez-Marroquin et al., 2021) can reveal the presence of biological species in an aerosol sample that has been found to contain INPs, it remains difficult to characterise the ice-nucleating ability of these species over other constituents (e.g. mineral dusts) when a sample's INA is analysed by, for example, a droplet freezing assay alone."

**R1C#12**. Following all these comments and suggestions, can the authors create a recommendation rubric for heat test measurements: type (wet vs dry), temperature, length of time, with the ultimate goal to streamline how our community runs these heat tests in the future (including with and without hydrogen peroxide as some groups have done.)

While developing a robust heat test protocol is the overarching aim, it is necessary to break this down into achievable steps. The first was to test the assumption that minerals are not sensitive to heat.  We have shown that they are and have provided the motivation to improve heat test protocols. At the end of Section 4 we detail improvements and caveats to using heat tests based on our findings.  For example we offer recommendations to adapt the wet-heat test to avoid mineral deactivations. ) and recommend control tests on known INP standards.. Also, as a result of the additional work that R1 has suggested we have concluded the following about dry heating:

**"Our data shows that dry heating at a lower temperature of 95 °C preserved the activity of K-feldspar, however, it did not deactivate heat-resistant polysaccharide INPs and denatured proteinaceous INPs (Snomax) far less than when heated at the same temperature in water. This means that, for detection of proteinaceous INPs, dry heating at 95 °C holds no advantage over wet heating at 95 °C."**

I'll just add a comment here to the authors, that I am very conscious of the additional efforts being requested for the revisions of this paper. Nonetheless, I think the authors and readers (including myself) would greatly benefit from the study being expanded in its conclusions and implications of the heat tests. I hope the authors will be encouraged to improve their work and go those extra steps further for the benefit of our community.

To summarise, we sadly did not have the time and resources to carry out all of the experiments suggested by R1 and we believe many of them are more appropriate for a follow up study. However, their comments were extremely helpful in expanding the conclusions and particularly for identifying the issue with the filtration equipment (R1C##).

---

## Author Comment (AC2)

Response to Reviewer 2 Comments.

We would like to thank the reviewer for their constructive feedback and suggestions. We present our responses and the resultant changes to the original manuscript below. We have re-numbered the reviewers' comments (R1 or R2 C#1, C#2 etc.) and split some up (a, b, c etc.) in order to respond to individual points. Our responses are written below in blue 10pt text with **changes indicated by bold type.**

**R2C#1**: 'Comment on amt-2021-208', Anonymous Referee #2, 11 Sep 2021
The authors investigated the responses of heat treatments (namely dry- and wet-) on different types of atmospheric ice-nucleating particle (INP) proxies using their offline cold stage instrument. Based on their findings, they made some technical recommendations regarding the offline heat treatment study of INPs (L695-725). The study objective and hypotheses are valid. The reviewer generally agrees that different ice-nucleating materials respond to heating in various ways (e.g., L576 etc.). The authors' messages are clear (L639-640; L643-645) while some explanations sound speculative. The reviewer has some major and minor comments. Some re-organizations of sections seem necessary to improve the readability.

Major comments

Proteinaceous structures can be destroyed below boiling temperature (Steinke et al., 2016). For example, Szyrmer and Zawadzki (1997) found some known cell-free IN-active microbes (e.g., Fusarium nuclei) are stable only up to 60 °C. Other studies of IN active bacteria, fungi, and lichens have shown heat sensitivity at lower than 100 oC. The reviewer is missing the detailed discussion of what protein is (and what is not) denatured in different temperature ranges. It is somewhat temperature-dependent and perhaps employing ~ 80 °C for 10-30 min (i.e., Fig. A2) may be comparable to using truly boiling temperature for heat treatment?

We agree that proteinaceous INPs can be denatured and deactivated at well below boiling temperature. We did already acknowledge that, for example, fungal INPs are known to have higher heat resistance compared to bacteria (lines 603-604 of original manuscript) and we have now elaborated on this. **It now states in Section 2.1.2 (Biogenic sample selection rationale) that 'T Fungal INPs, have been found to have slightly higher heat resistance in wet mode compared to bacterial INPs, typically showing no reduction in INA with up to 60 °C of heating compared to 40 °C with bacterial INP but for both these it is eliminated by heating above 90 ° C. (Pummer et al., 2015; Pouleur et al., 1992; Fröhlich-Nowoisky et al., 2015),**

The motivation of using 100 °C (or near 100 °C) as the temperature for the wet heat treatment is primarily that maintaining a water bath at boiling temperature is more practically convenient than maintaining one at 60 or 80 for example, hence it is commonly used in the community. Another reason for using these conditions was that they are the representative of those used by previous workers performing heat tests on samples taken from the field.

**R2C#1a**. What is the minimum time for proteins to be denatured?

To section 2.3.1 (Wet heating methodology) we added:

**"Samples of proteinaceous IN derived from lichen (Kieft 1988, Kieft and Ruscetti 1990, Moffat et al, 2015), fusarium fungi (Poleur et al. 1992) and psuedomonas syringae bacteria (Maki et al. 1974) all saw large deactivations after being heated in boiling water baths for less than 15 minutes, presumably due to denaturation. Therefore, it is presumed that 30 min of immersion under these conditions is sufficient to denature INP of proteinaceous origin."**

**R2C#2**. Fig. 1 and all associated discussions fit better in the results & discussion section rather than the materials & method section.

This figure has been moved to Section 3 (Results and discussion) prior to the sub-sections for each INP classes' results.

**R2C#3**. L338-358 & L466-467 & L478-479: Hydrolysis and dry-heating likely alter SSA and other physical properties of materials, which impact their ice nucleation abilities. Reporting SSAs of a subset of materials after wet- and dry-heating would clarify the authors' hypothesis given in these parts and strengthen the paper. Unless the authors can directly quantify the loss of active sites and/or the number of denatured proteins by a set of heat treatments, some arguments seemingly remain speculative

We reiterate the main focus of this paper was to empirically determine the sensitivity of ice-nucleating materials to heat, to test commonly used heat tests. That said, we also felt that the results have some fundamental mechanistic value and included the discussion accordingly. Our work clearly indicates future areas for research.

Regarding SSA measurements, we do not think a change in INA of minerals would necessarily correlate with a change in its total surface area after a heat treatment. We used a singular (active site) approach to conceptualise ice nucleation and there is physical evidence that immersion mode ice nucleation activity on quartz and K-feldspar occurs at specific active sites (Holden et al 2019). Therefore, the INA of these samples is contained within a very small proportion of the total surface area of the sample and any treatment that 'destroys' the active sites could be possible without a detectable change of SSA.

**R2C#4**. Sect. 2.3.2: The choice of 250 dC for dry-heating seems appropriate, but if the authors wish to do the apples-to-apples comparison of wet-heating vs. dry-heating, wouldn't it make more sense to use the same heating temperature and period for both heating methods? The authors state that "The dry heat test is a harsher treatment than wet heating…" in L369-370. Do the authors think the measurements with multi-temperatures could be a better procedure in dry heat tests (e.g., 100 dC vs. 250 dC etc.)? Perhaps, Amazonite microcline may have a different response to dry heat at 100 dC? SSA may be changing depending on the employed heat temperature?

Please refer to RC1## with regards to heating conditions and RC2## with regards to surface area changes.

**R2C#5**. The snapshot example of quartz in Fig. A1 is very nice. The reviewer wishes to see a similar dataset for wet-heat stable compounds (e.g., kaolinite and MCC). Does a similar trend hold for non-quartz samples?

Fig A1 shows the effect of heating Fluka Quartz, a wet heat sensitive sample, in different types of containers made of glass and plastic. This extra set of experiments was specific to quartz due to the

findings of Kumar et al., 2019 who found quartz INP suspensions may interact with glass containers in a way that reduces their INA, but not plastic containers. In light of this remarkable finding we wanted to eliminate this as the cause of the wet-heat deactivations seen with the quartz samples. Doing similar experiments with other types of INP would indeed be an interesting future study, but there is no indication from our or other people's experiments that there is a dependency on vessel type for other materials.

**R2C#6**. Sect. 2.2.: The used suspension concentration of 1% w/v for MCC etc. seems to exceed what is recommended in previous literature (e.g., Sect. 3.1. in Hiranuma et al., 2019). What is the rationale behind such a high concentration? Wouldn't such a high concentration cause some issues (e.g., flocculation of suspended particles)?

Please refer to RC1#6a where we addressed this issue. **We repeated experiments with MCC with lower concentrations of 0.1% and 0.01%. When plotted as ns(T) plots (Fig B1d), both runs agree well with the wet-dispersion parameterisation from Hiranuma et al 2015 which we have plotted**. Also, the results, which used the same protocol, agree with the μl-NIPI data in from Hiranuma et al 2019.

**R2C#7**. How do these high concentration ns spectra compare to previous studies? The reviewer sees the K-feldspar reference spectrum in Fig. 2 but not for other materials the authors examined for this study.

**To the $n_s$(T) plots in Appendix B, Fig B1 we have added parameterisation spectra for quartz, Snomax, BPWW and MCC**. Of these, Fluka Quartz at high concentration of 2.5% w/v is the only sample that appears to deviate from previous studies (Harrison et al., 2019) being about an order of magnitude higher, although this parameterisation not based on this specific quartz sample, rather on an amalgamation of several silica INP samples.

**R2C#8**. L648-670: Indistinguishable by the heat reaction itself but complementary mineralogy and composition analyses can distinguish these two populations.

We agree and have added **"Interpretation of results may by aided by identification of the mineral phases present in a sample using techniques such as XRD or SEM" in the 3rd paragraph of section 5 (Conclusions).**

**R2C#9**. Do the authors intend to argue the applicability of heating on environmental samples (i.e., the mixture of different compositions)?

Section 4 contains discussion of this issue, for example:

"Generally, marginal heat deactivations should be interpreted with caution and generally should not be attributed to the presence of proteinaceous ice-nucleating materials. This especially applies if heat deactivations have been used to calculate the ambient concentration of biological INPs in addition to identifying their presence."

**R2C#10**. L755-756: The reviewer thinks that online heating (i.e., INP measurement with a heating inlet etc.) could be a good alternative approach for the quantitative test. Perhaps, the offline heating tests can be done with a set of different temperatures? Include these points in P25 (a), (b), and (d)?

Online heating with a heated inlet would indeed be a valuable experimental technique, but we simply cannot access the short time scales (that the aerosol would spend in an inlet) with our set up. To explore this we would need to build a dedicated system.

Minor comments

**R2C#11**. What is the minimum detection limit of evaluation ice nucleation ability and/or efficiency of the cold stage for this study?

Data for blank runs (and handling blanks) are shown in each boxplot fig and in supplementary figure S3 and show $T_{50}$ values that ranged from -27.4 to -25.6 °C. This is typical of background runs using 1 µL ultrapure water droplets using the µL-NIPI instrument (Whale et al., 2015). Adding the significance threshold of +/- 1.2 °C we introduce in section to the warmest background $T_{50}$ value we can say that any sample run with a $T_{50}$ value warmer than -24.4 °C is deemed as significantly above the limit of detection. All mineral samples, heated and unheated, fell above this threshold in terms of $T_{50}$. Previous studies which have used the µl-NIPI instrument have used a more sophisticated background subtraction routine to determine whether signals are above detection limits (Umo et al. 2015, Wilson et al 2015). However, a relatively simple approach is justified here because of the relatively high concentrations of INPs we used, which ensured most runs produced freezing spectra well above blank levels.

**R2C#12**. L25-28: This sentence makes sense without the last few words (", so long as … K-feldspar"). This part sounds speculative. The reviewer suggests removing this part from the abstract. The importance of K-feldspar as an INP seems not the main focus of this study.

Reworded to the following at the end of the Abstract:

"We conclude that, while wet INP heat tests at (>90 °C) have the potential to produce false positives, i.e., deactivation of a mineral INA that could be misconstrued as the presence of biogenic INPs, they are still a valid method for qualitatively detecting proteinaceous biogenic INP in ambient samples **if** the mineral-based INA is controlled by K-feldspar. "

**R2C#13**. L57-59: Adding a discussion of the emission rates and atmospheric abundances of mineral dust and biogenic INPs might make this paragraph even more meaningful. Please consider providing some information to the reader.

We have added some extra background info on mineral and biogenic particle abundances in the 4[th] paragraph of the introduction

**Atmospheric concentrations of ice-active bacteria, fungal spores and pollen grains are much smaller than mineral dusts (Hoose et al., 2010). Estimates of the mass of PBAPs emitted to the atmosphere anuallyannually range from low hundreds to ~1000 Tg (Hoose et al., 2010; Jaenicke et al 2005) compared to 1,000 - 3,000 Tg per year for mineral dust (Zender et al., 2004). However, the concentration of fragments of biogenic INPs may be much greater given the release of macromolecular INPs (Augustin et al., 2013; O′Sullivan et al., 2015) and their adsorption onto lofted soil dust (Schnell and Vali, 1976; O'Sullivan et al., 2016). Also the sources and atmospheric distribution of biogenic INPs are less well characterised compared to those of minerals dusts (Huang et al., 2021; Kanji et al., 2017), owing to the diversity of marine and terrestrial sources that may be subject to seasonal variations (Conen et al., 2015; Schneider et al., 2020; Šantl-Temkiv et al., 2019) or influenced by anthropogenic activities such as agricultural processes (Garcia et al., 2012; Suski et al., 2018; O'Sullivan et al., 2014).**

**R2C#14**. L67-70: Tobo et al. (2019) shows soil dust has some contributions to it, too. This can be briefly discussed here?

We acknowledge this by adding the citation to the introduction, 5th paragraph:

"Arctic climate, as increasing surface temperatures may expose new terrestrial sources in thawing permafrost (Creamean et al., 2020), **newly exposed glacial outwash sediments (Tobo et al., 2019)"**

We also mention studies which inferred the presence of biological INP associated with soil dusts in the paragraph above:

"However the abundance of biogenic INPs of may be much greater than estimates of PBAP abundances suggest given the release of macromolecular INPs (Augustin et al., 2013; O'Sullivan et al., 2015) and their adsorption onto lofted soil dust (Schnell and Vali, 1976; O'Sullivan et al., 2016)".

**R2C#15**. L287-293: Repetitive, and this part does not fit in the results and discussion section.

We included quite a lot of background material on the mineral samples we used and agree they do not belong in the results section, specifically reasons for including each individual sample, general properties and atmospheric relevance and literature references to past INA measurements. **Therefore, we have moved this background material to new sections:**
- **2.1.1 Mineral sample selection rationale**
- **2.1.2 Biogenic sample selection rationale**
- **Supplementary section S1: Background information on classes of mineral INP**

This has considerably reduced the amount of text in the results section and improved readability.

**R2C#16**. L295-303: Better fits in the materials & method section.

See response to comment R2C15 above

**R2C#17**. L307-307 & L316-318: The authors can introduce a brief statement to guide the reader where the explanation of the observed results is given later in this section (L349-).

Added "**(we discuss the possible reasons for this later in this section) in the first paragraph of Section 3.1.1.**

**Removed the sentence on lines 316-318 as this is just repeating the previous sentence. It now flows straight into the explanation of the results.**

**R2C#18**. L326-336: Fits better in the intro section.

See response to comment R2C15 above

**R2C#19**. L399-403: Fits better in the materials and methods section.

See response to comment R2C15 above

**R2C#20**. L405: Quartz presented before feldspar in Fig. 3a. Fig. 3b is not discussed until Page 17. The arrangement of Fig. and sub-sections seems a bit odd.

We have re-organised the figures as detailed above, including splitting up the boxplots with silica and plagioclase samples.

**R2C#21**. L430-431: Sounds speculative.

**We agree with this since we are effectively extrapolating our results to those of Amelia Albite (Harrison et al., 2016) which is an atypical sample of plagioclase feldspar. We have therefore removed Lines 429-431.**

**R2C#22**. L433-455: Fits better in the materials and methods.

See response to comment R2C#15 above

**R2C#23**. L504-521: Fits better in the materials and method section or SI.
L549-558: Fits better in the materials and method section or SI
L596-598: Fits better in the materials and method section or SI.

See response to comment R2C#15 above

**R2C#24**. L684-691: Sounds speculative

We disagree that this is speculative as there is enough evidence in the literature that grinding and milling increases the INA of some minerals (in the 5[th] paragraph of Section 4 Summary and implications for using INP heat treatments)): ". The INA of quartz (Kumar et al., 2019a; Zolles et al., 2015; Harrison et al., 2019), hematite (Hiranuma et al., 2014) and also natural desert dusts (Boose et al., 2016) are increased by milling". Moreover, it is important that there also be some discussion about the relevance of results seen in the laboratory to real processes in the environment.

**R2C#25**. L739-741: Showing the altered specific surface after wet heating can be more direct evidence. Accounting a different SSA may explain the alternation in IN ability of the material?

See response to comment R2C#3 above

R2C#26. Fig. S1: The axis text/numbers and legends are too small to see.

**Reorganised so is now visible.**

**R2C#27**. Fig. S3: Is this ns plot generated using the FF spectra data in Fig. S1(u)? It seems that Fig. S3 is missing the point above -12 dC for a non-treated sample. Or the authors did another set of measurements for generating Fig. S3? Please clarify.

The data point above –12 °C is part of the dry-heated sample and not included on Figure S3 (which is now moved to Figure 4b).

Technical comments

L3: Murray11

Done

L30: à the absence of ice nucleation active sites,

Done

L39 our models à atmospheric models

Done

L56: 2015b comes before 2015a

Done

L63: ice active à ice nucleation active

Done

L63-66: How small & how great? The reviewer suggests that the authors provide some quantitative information to the reader in this paragraph.

See response to R2C#13**, we have added quantitative info.**

L82: à characterize the ice-nucleating activity of

Done

L84: à known bacterial, fungal, and archaeal

Done

L250-274: "A" should be defined in L251, or the authors can introduce Eqn. 2 in L273.

**In Section 2.4 (Ice nucleation measurements...) we have removed this equation and replaced with three separate ones (Eqns 1-3) which define $f_{ice}(T)$   $n_s(T)$   and $n_m(T)$.**

L306: after a closing-parenthesis change "," to ".".

Done

L409: ))

Done

L770: 1 mL or 1.5 mL? One figure says 1.5. What is the vendor and model number of the tube?

**1.5 mL (Sarstedt Micro Tube 72.690) – added to Appendix A.**

References:

[revised manuscript text omitted]

---

## Author Response (AR2)

Dear Editor

Please find our responses to the referee comment reports below. The comments in Report #1 resulted in us editing the terminology used for our biological INP samples on the basis of their heat-resistance rather than composition. We appreciate their valid suggestions of alternative treatments such as autoclaving and denaturant treatment. However, we reiterate that the motivation of the study is to only assess treatments that involve heat for biological INP detection. In response to Report #2 we updated several figures to include handling blank data, discussed the filtration contamination issue and reported the data in the Supplementary Information.

Other than sections identified below where discussions have been added and minor edits made to text throughout the manuscript, no other substantial changes have been made. We sincerely thank the anonymous reviewers for their scrutiny and useful suggestions and we hope these improvements to the paper address their concerns.

***We have re-numbered the reviewers' comments (Comment#1, Comment#2 etc.) and split some up (a, b, c etc.) in order to respond to individual points. Our author responses are written below in blue 11pt text.***

Response to Report #1 (anonymous referee #3)

The authors compare dry and wet heating of ice nucleating particles (INP) and ice nucleating macromolecules (INM), some of which are biological or were eventually in contact with biological material. I agree with the general conclusion of the authors concerning inappropriate differentiation between inorganic and biological material by heating tests. I also agree that the manuscript has significantly been enhanced after the first round of discussion with two anonymous referees. Nevertheless, I would like to add some comments and concerns, which the authors might consider.

Comment #1a: I am not a biochemist, but I have learnt so far that dry heating is not a suitable way to denature proteins. The appropriate way is wet heating in an autoclave, which allows temperatures much higher than 100°C without falling dry of the sample. Otherwise, without water the proteins aggregate and can encapsulate and their structures can partially survive which might not be enough for biological activity but is enough at least for ice nucleation. I strongly encourage the authors to use or at least to recommend an autoclave for heating experiments and also to use other tests than only heating, e.g., incubation with subtilisin and urea, which is unfolding the protein structure and have proven the erasure of ice nucleation activity.

We have focussed the study solely on INP sample treatments that involve heat and rather than the use of chemical additives. Using autoclaving is, however, certainly a valid alternative method to 'wet' and 'heating' tests, at least for practical reasons as it does not involve adding chemicals to a sample and portable autoclaves could reasonably be deployed on field campaigns. We note that autoclaves typically use saturated steam at a temperature of about 120°C. This is somewhat harsher than our wet heating and before recommending this as a treatment we would need do a set of tests with all material tested here. We have already found, in some preliminary tests, that the activity of K-feldspar is strongly reduced when autoclaved, hence we suspect that it may not be useful in identifying biological and mineral ice nucleators. Conversely, dry heat offers the potential for separation of biological and mineral ice nucleators.

Comment #1b: The authors might also read and quote the respective literature form the fields of biochemistry.

To provide some extra context to the results we saw with dry heating Snomax at 'low' temperature we have added a discussion about the effect of water content on protein denaturing temperature with reference to the literature at the end of Section 3.2.

Major comment

Comment #2: My own experience with birch pollen washing water was that I was originally convinced that proteins can be excluded as INMs due to heating tests (Pummer et al. 2012), but only recently fluorescence spectroscopy and incubation tests proofed, that proteins are certainly involved in the ice nucleating mechanism (Burkart et al. 2021). Therefore, I would be extremely careful with statements correlating heating treatment tests with protein content if not other reliable methods have been chosen as well. In the data of this manuscript a downshift of the nucleation temperature due to heating of birch pollen washing water is indeed visible (1 °C), which however is small in comparison to Snomax samples.

We appreciate this being pointed out and in light of this we changed our classification of the biological INP samples from 'proteinaceous' and 'non-proteinaceous/polysaccharide' to 'heat-sensitive' and 'heat-resistant'. We also updated our description of the nature of birch pollen INP in Section 2.1.2 with appropriate references added and we also removed references throughout to pollen INP explicitly being polysaccharide based.

Minor comment

Comment #3 : In Table 1 the data set of Seifried et al., 2021 is missing, where heating of a bioaerosol sample at 98 °C for a duration of 1 hour has been investigated.

This paper has been added to Table 1.

Response to Report #2 (anonymous referee #1)

Comment #1: The authors substantially improved their manuscript and the quality of their data. I am concerned that the originally omitted handling blanks revealed contamination issues. The authors repeated a subset of their experiments for the birth pollen and noticed that in fact, birch pollen did not nucleate ice above their background after heating (consistent with (Pummer et al., 2012)). When I compare the first Figure 1 and the revised Figure 1, the changes are substantial, and in my opinion, require a discussion of contamination explicitly in the main manuscript. I am concerned that the authors replaced the data without any comments about how problematic contamination of their filter holder was. Let's make sure we let the community know the importance of handling blanks in ice nucleation, and that as a community we can improve in our analytical approach.

We added a short discussion in Section 2.2 to address this and also added the data showing the contamination from the stainless steel filter holder to Fig S3.

Here are my additional comments:

Comment #2: - Define blanks in Figure 1 – handling blanks that went through the setup should be used when possible.

To this figure we added the handling blanks for wet and dry heating and more clearly defined the blanks #1-3 as 'clean water blanks'. Blanks #4-5 were removed for clarity but remain in Fig S3.

Comment #3: - Figure 2 is improved, the blanks as box plots at the bottom are important to include. Define blank runs explicitly (through filter, through glassware, etc.) And rather than include #1-#5, the handling blanks can also be included.

The handling blanks and clean water blank data in Fig 1 have been added to Figs 2-7 in boxplot form. In addition, the filtration handling blank data has been added to Fig 7.

Comment #4:- Figure 4 – plot T50 as function of time immersed in solution

This has been added as Fig 4c.

Comment #5:- Again, to reiterate: I encourage the authors to be transparent about their discovery of the stainless steel filter contamination and the importance of handling blanks. I very much appreciate figure S3. And I recommend that the authors discuss the observation of contamination that led to substantial differences from the original Figure 1.

See reply to Comment #1.

Comment #6: On a more philosophical point, as scientists we have the responsibility to be our own harshest critics. If additional experiments are needed to clarify ambiguities and speculations, then there really shouldn't be pushback on validating our experiments. I know the work is hard and long, but I would argue that's what we signed up for as scientists: to be rigorous and trustworthy. Food for thought when addressing the following comment made by the authors: "To summarise, we sadly did not have the time and resources to carry out all of the experiments suggested by R1 and we believe many of them are more appropriate for a follow up study. However, their comments were extremely helpful in expanding the conclusions and particularly for identifying the issue with the filtration equipment (R1C##)."

We simply have to disagree on this point. There are limits to how much time people can spend on a project. We have made an important step forward in this paper and it opens up a great deal of future research on several fronts that should be tackled in the future. Also, this is already a long paper with lots of data.